

# Air-sea momentum flux climatologies: A review of drag relation for parameterization choice on wind stress in the North Atlantic and the European Arctic

Iwona Wrobel-Niedzwiecka, Violetta Drozdowska, Jacek Piskozub[1]

[1] Institute of Oceanology, Polish Academy of Science, ul. Powstancow Warszawy 55, 81-712 Sopot, Poland
corresponding author: iwrobel@iopan.gda.pl

Key points: drag coefficient, European Arctic, North Atlantic, parameterizations

## 1 Abstract

In this paper we have chosen to check the differences between the relevant or most commonly used parameterizations for drag coefficient ($C_D$) for the momentum transfer values, especially in the North Atlantic (NA) and the European Arctic (EA). As is well know, the exact equation in the North equation that describes the connection betwenn the drag coefficient and wind speed depends on the author. We studied monthly values of air-sea momentum flux resulting from the choice of different drag coefficient parameterizations, adapted them to momentum flux (wind stress) calculations using SAR wind fields, sea-ice masks, as well as integrating procedures. We calculated monthly momentum flux averages on a 1º x 1º degree grid and derive average values for the North Atlantic and the European Arctic. We compared the resulting spreads in momentum flux to global values and values in the tropics, an area of prevailing low winds. We show that the choice of drag coefficient parameterization can lead to significant differences in resultant momentum flux (or wind stress) values. We found that the spread of results stemming from the choice of drag coefficient parameterization was 14 % in the Arctic, the North Atlantic and globally, but it was higher (19 %) in the tropics. On monthly time scales, the differences were larger at up to 29 % in the North Atlantic and 36 % in the European Arctic (in months of low winds) and even 50 % locally (the area west of Spitsbergen). When we chose the oldest parameterization (e.g Wu, 1969 (W69)) values of momentum flux were largest for all months, in compare to values from the two newest parameterizations (Large and Yeager, 2004 (LY04) and Andreas, 2012 (A12)), in both regions with high and low winds and $C_D$ values were consistently higher for all wind speeds. For global data not much seasonal change was note due to the fact that the strongest winds are in autumn and winter as these seasons are inverse by six months for the northern and southern hemispheres. The situation was more complicated when we considered results from the North Atlantic, as the seasonal variation in wind speed is clearly marked out there. With high winter winds, the A12 parameterization was no longer the one that produces the smallest wind stress. In this region, in summer, the highest wind stress values were produced by the NCEP/NCAR reanalysis, where in $C_D$ has a constant value. However, for low summer winds, it is the lowermost outlier. As the A12 parameterization behaves so distinctly differently with low winds, we showed seasonal results for the tropical ocean. The sequence of values for the parameterization was similar to that of the global ocean, but with visible differences betwenn NCEP/NCAR, A12 and LY04 parameterizaions. Because parameterization is supported with the largest experimental data set observations of very low (or even negative) momentum flux



values for developed swell and low winds, our results suggest that most circulation models
overestimate momentum flux.

## 1. Introduction

Wind stress acts at the air-sea interface influence on wind-wave interaction, including
wind-driven surface waves, turbulence in upper and deep layers, drift currents, and the main
ocean currents (Zilitinkiewicz et al., 1978). The ocean surface mixed layer is a region where
kinematic forcing affects the exchange of horizontal momentum and controls transport from
the surface to depths (Gerbi et al., 2008, Bigdeli et al., 2017). Any attempt to properly model
the momentum flux from one fluid to another as the drag force per unit area at the sea surface
(surface shear stress, $\tau$) must take into account other physical processes responsible for
generating turbulence such as boundary stress, boundary buoyancy flux, and wave breaking
(Rieder et al., 1994, Jones and Toba, 2001). Over the past fifty years, as the entirety of flux
data has increased many fold, multiple empirical formulas have been developed to express the
ocean surface momentum flux as a relationship between non-dimensional drag coefficient
($C_D$), wind speed ($U_{10}$), and surface roughness ($z_0$) (Wu 1969, 1982; Bunker, 1976; Garratt,
1977; Large and Pond, 1981; Trenberth et al., 1989; Yelland and Taylor, 1996, Donelan et al.,
1997; Kukulka et al., 2007; Andreas et al., 2012). These formulas can be divided into two
groups. One group of theories gives the $C_D$ at level $z$ in terms of wind speed and possibly one
or more sea-state parameters (for example, Geernaert et al., 1987, Yelland and Taylor, 1996,
Enriquez and Friehe, 1997), while the second group provides formulas for roughness length $z_0$
in terms of atmospheric and sea-state parameters (for example, Wu, 1969, Donelan et al.,
1997, Andreas et al., 2012). It is well known that the drag coefficient is not a constant,
because surface roughness changes with sea state, and that it is an increasing function of wind
speed for moderate wind speeds in the marine atmospheric boundary layer (Foreman and
Emeis, 2010). On the other hand, many researchers have recently shown that results for the
drag coefficient are underestimated at moderate wind speeds and overestimated at high wind
speeds (Jarosz et al., 2007, Sahlée et al., 2012, Peng and Li, 2015, Brodeau et al., 2017).

In this paper we chose to check the differences between the relevant or most
commonly used parameterizations for drag coefficient ($C_D$) for momentum transfer values,
especially in the North Atlantic (NA) and the European Arctic (EA). As is widely known, the
exact equation that describes the connection between the drag coefficient and wind speed
depends on the author (Geernaert, 1990). Our intention here is not to re-invent or formulate a
new drag parameterization for the NA or the EA, but to revisit the existing definition of drag
parameterization, and, using satellite data, to investigate how existing formulas accommodate
the environment in the North. We concentrated on wind speed parameterizations, because
wind speed is a parameter that is available in every atmospheric circulation model. Therefore,
it is used in all air-sea flux parameterizations, and presently it is used even when sea state
provides a closer physical coupling to the drag coefficient (for review see Geernaert et al.,
1986).




To understand air-sea interaction, Taylor (1916) parameterized the wind's drag on the
sea surface using the bulk aerodynamic formula:

$$\tau = \rho C_{Dz} U_z^2 \tag{1}$$

where ($\tau$) is the drag per unit area of sea surface (also called surface stress or momentum flux),
$\rho$ is air density, $C_{Dz}$ is the non-dimensional drag coefficient appropriate for $z$ height, and $U_z$ is
the average wind speed at some reference height $z$ above the sea. $C_{Dz}$ is commonly
parameterized as a function of mean wind speed (m s$^{-1}$) for neutral-stability at a 10 m
reference height above mean sea level (Jones and Toba, 2001), which is identified as $C_{DN10}$ or
$C_{D10}$ (this permits avoiding deviation for the vertical flow from the logarithmic law):

$$C_{DN10} = \frac{\tau}{\rho} \, U_{10}^2 = \left(\frac{u_*}{U_{10}}\right)^2 \tag{2}$$

where $u_*$ is friction velocity. Alternatively, the neutrally stratified momentum flux can be
determined from the logarithmic profile, thus Eq. 1 can be express as:

$$C_{DN10} = [\kappa / ln \, (10/z_0)]^2 \tag{3}$$

where $z_0$ (m) is the aerodynamic roughness length, which is the height, above the surface to
define the measure of drag at which wind speed extrapolates to 0 on the logarithmic wind
profile (Andreas et al., 2012), and $\kappa$ is von Kármán constant ($\kappa$=0.4).

At the same time, we can define the value of friction velocity ($u_*$) as having the
dimension of velocity, which is defined by the following equation:

$$\tau = \rho \, u_*^2 \tag{4}$$

Comparison with bulk formula (1) leads to the equation:

$$u_*^2 = C_{D10} U_{10}^2 \tag{5}$$

Some of the first studies (Wu, 1969, 1982, Garrat, 1977) focused on the relationship
between wind stress and sea surface roughness, as proposed by Charnock (1955), and they
formulated (for winds below 15 m s$^{-1}$) the logarithmic dependence of the stress coefficient on
wind velocity (measured at a certain height) and the von Kármán constant. Currently common
parameterizations of the drag coefficient are a linear function of 10 m wind speed ($U_{10}$), and
the parameters in the equation are determined empirically by fitting observational data to a
curve. The general form is expressed as (Guan and Xie, 2004):
$$C_D 10^3 = (a + bU_{10}) \tag{6}$$
Wu (1969), based on data compiled from 12 laboratory studies and 30 oceanic
observations, formulated power-law (for breezes and light winds) and linear-law (for strong
winds) relationships between the wind-stress coefficient ($C_y$) and wind velocity ($U_{10}$) at a
certain height $y$ at various sea states. In his study, he used roughness Reynolds numbers to





characterize the boundary layer flow conditions, and he assumed that the sea surface is
aerodynamically smooth in the range of $U_{10} < 3$ m s$^{-1}$, transient at wind speed $3$ m s$^{-1} < U_{10} <$
$7$ m s$^{-1}$, and aerodynamically rough at strong winds $U_{10} > 7$ m s$^{-1}$. He also showed that the
wind-stress coefficient and surface roughness increase with wind speed at light winds ($U_{10} <$
$15$ m s$^{-1}$) and is constant at high winds ($U_{10} > 15$ m s$^{-1}$) with aerodynamically rough flow.
Garratt (1977), who assessed the 10 m neutral drag coefficient ($C_{DN10}$) based on 17
publications, confirmed the previous relationship and simultaneously suggested a linear form
of this relationship for light wind. Wu (1980) proposed the linear-law formula for all wind
velocities and later (Wu, 1982) extended this even to hurricane wind speeds. All of the
preceding results rely heavily on the Monin-Obukhov similarity theory (MOST) in order to
eliminate the stability dependence by choosing 10 m as the standard reference height and
using data obtained under different experimental conditions (laboratory or field) and data
analysis. In 1981, Large and Pond's estimated momentum flux using the direct Reynolds flux
method and the dissipation method indicated the linear-law of $C_D$ for wind speed in moderate
winds. Their results confirm the assumption that the neutral drag coefficient increases with
higher wind speed values and support the theoretical prediction that $C_{DN}$ is independent of the
bulk stability parameter (z/L). Trenberth et al. (1989), who considered the uncertainty in $C_D$
from earlier experiments in which there were difficulties in calculations for low frequency
(less than 10 days), suggest incorporating a quantity called pseudostress (P), which assumes
using an effective drag coefficient and constant air density. Their results were based on data
from the European Centre for Medium Range Weather Forecasts (ECMWF) collected over
seven years. Yelland and Taylor (1996) presented results obtained from three cruises using
the inertial dissipation method in the Southern Ocean and indicate that using the linear-law
relationship between the drag coefficient and wind speed (for $U_{10} > 6$ m s$^{-1}$) is better than
using friction velocities ($u_*$) with $U_{10}$. Fairall et al. (2003) used the COARE algorithm
(Coupled Ocean-Atmosphere Response Experiment) globally as a function of ambient
conditions. Their results with direct covariance flux measurements showed increases in $C_{DN10}$
values from $1.0 \times 10^{-3}$ to $2.3 \times 10^{-3}$ (or to $2.07 \times 10^{-3}$ if inertial dissipation fluxes were used)
with increasing wind speed (from $3$ m s$^{-1}$ to $20$ m s$^{-1}$). All of these studies show that
coefficients are not identical and vary with wind speed and atmospheric stability.
Authors of coupled circulation models preferred even simpler parameterizations. The
NCEP/ NCAR reanalysis (Kalnay et al., 1996) uses a constant drag coefficient of $1.3 \times 10^{-3}$
while, for example, the Community Climate System Model version 3 (Collins et al., 2006)
uses a single mathematical formula proposed by Large and Yeager (2004) for all wind speeds.
Their parameterizations explicitly or implicitly assume that equation (6) is exact. However,
Foreman and Emeis (2010) show that friction velocity is proportional to wind speed, but with
offset:
$$u_* = aU_{10}^2 + b \tag{7}$$
Andreas et al. (2012), further referred to as A12, updated equation (8) based on available
datasets, friction velocity coefficient ($u_*$) versus neutral-stability wind speed at 10 m ($U_{N10}$),
and sea surface roughness ($z_0$), to find the best fit for parameters $a = 0.0583$ and $b = -0.243$.





They justify their choice by demonstrating that $u_*$ vs. $U_{N10}$ has smaller experimental
uncertainty than $C_{DN10}$, and that one expression of $C_{DN10}$ for all wind speeds overstates and
overestimates results in low and high winds (**Figs. 7** and **8** in A12).
This led directly to a new $C_D$ formulation with much lower values for light winds (4 -
9 m s$^{-1}$). These low values could explain why the observed momentum flux with light winds
and fast traveling swell can even be negative (Grachev and Fairall, 2001; Hanley and Belcher,
2008), and if true, this means that all previous parameterizations overestimate wind stress in
basins with prevailing light winds (for example, the tropics).
All the above studies propose different parameterizations (see **Fig. 1**) of the drag
coefficient and the function of wind speed, which reflects the difficulties in simultaneously
measuring at hight sea stress (or friction velocity) and wind speed. The purpose of this study
is to show how the choice of $C_D$ parameterization influences the value of the momentum flux
from the atmosphere to the ocean with observations based on wind fields in different parts of
the ocean, but especially in the NA and the EA seas.

## 2. Materials and Methods

We calculated monthly and annual mean momentum fluxes using a set of software
processing tools called FluxEngine (Shutler et al., 2016), which was created as part of the
OceanFlux Greenhouse Gases project funded by the European Space Agency (ESA). Since
the toolbox, for now, is designed to calculate only air-sea gas fluxes but it does contain the
necessary datasets for other fluxes, we made minor changes in the source code by adding
parameterizations for the air-sea drag relationship. For the calculations, we used Earth
Observation (EO) wind speed data at 10 m above sea level for 1992-2010 and sea roughness
(σ0 – altimeter backscatter signal in the Ku band) from the GlobWave project
(http://globwave.ifremer.fr/). GlobWave produced a 20-year time series of global coverage
multi-sensor cross-calibrated wave and wind data, which are publicly available at the
Ifremer/CERSAT cloud. Satellite scatterometer derived wind fields are at present believed to
be at least equally as good as wind products from reanalyses (see, for example, Dukhovskoy
et al. 2017) for the area of our interest in the present study. The scatterometer derived wind
values are calibrated to the equivalent neutral-stability wind at a reference height of 10 m
above the sea surface, and, therefore, are fit for use with the neutral-stability drag coefficient
(Chelton and Freilich, 2005). Wave data were collected from six altimeter missions (like
Topex/POSEIDON, Jason-1/22, CryoSAT, etc.) and from ESA Synthetic Aperture Radar
(SAR) missions (ERS-1/2 and ENVISAT). All data came in netCDF-4 format. The output
data is a compilation file that contains data layers, and process indicator layers. The data
layers within each output file include statistics of the input datasets (e.g., variance of wind
speed, percentage of ice cover), while the process indicator layers include fixed masks as land,
open ocean, coastal classification, and ice.
All analyses using the global data contained in the FluxEngine software produced a
gridded (1° x 1°) product. The NA was defined as all sea areas in the Atlantic sector north of
30° N, and the EA subset was those sea areas north of 64° N (**Fig. 6**). We also defined the





subset of the EA east of Svalbard ("West Svalbard" between 76º and 80º N and 10º to 16º E),
because it is a region that is studied intensively by multiple, annual oceanographic ship
deployments (including that of the R/V Oceania, the ship of the institution the authors are
affiliated with). FluxEngine treats areas with sea-ice presence in a way that is compatible with
Lüpkes et al. (2012) multiplying the water drag coefficient by the ice-free fraction of each
grid element. We also define "tropical ocean" as all areas within the Tropics (23º S to 23º N,
not show) in order to test the hypothesis that the new A12 parameterization will produce
significantly lower wind stress values in the region.
In this study, we calculated air-sea momentum flux average values using seven
different drag coefficient parameterizations ($C_D$). All of them are generated from the vertical
wind profile, but they differ in the formulas used.
$$10^3 \cdot C_{D10} = 0.5U_{10}^{0.5} \qquad \text{for } 1 \text{ m s}^{-1} < U_{10} < 15 \text{ m s}^{-1} \qquad (8)$$
(Wu, 1969)
$$10^3 \cdot C_{DN10} = 0.75 + 0.067U_{10} \qquad \text{for } 4 \text{ m s}^{-1} < U < 21 \text{ m s}^{-1} \qquad (9)$$
(Garratt, 1977)
$$10^3 \cdot C_{D10} = (0.8 + 0.065U_{10}) \qquad \text{for } U_{10} > 1 \text{ m s}^{-1} \qquad (10)$$
(Wu, 1982)
$$10^3 \cdot C_{DN10} = 0.29 + \frac{3.1}{U_{10N}} + \frac{7.7}{U_{10N}^2} \qquad \text{for } 3 \text{ m s}^{-1} < U_{10N} < 6 \text{ m s}^{-1} \qquad (11)$$
$$10^3 \cdot C_{DN10} = 0.60 + 0.070U_{10N} \qquad \text{for } 6 \text{ m s}^{-1} < U_{10N} < 26 \text{ m s}^{-1}$$
(Yelland and Taylor, 1996)
$$10^3 \cdot C_D = 1.3 \qquad \text{everywhere} \qquad (12)$$
(NCEP/NCAR)
$$10^3 \cdot C_{DN10} = \frac{2.7}{U_{10N}} + 0.142 + 0.076U_{10N} \qquad \text{everywhere} \qquad (13)$$
(Large and Yeager, 2004)
$$C_{DN10} = \frac{u*}{U_{10N}} = a^2 \left(1 + \frac{b}{a} U_{10N}^2\right) \qquad \text{everywhere} \qquad (14)$$
where $\quad a = 0.0583, b = -0.243$ (Andreas et al., 2012)
where $C_{DN10}$ is the expression of neutral-stability (10-m drag coefficient), $C_{D10}$ is the drag
coefficient dependent on surface roughness, $U_{10}$ is the mean wind speed measured at 10 m
above the mean sea surface, $U_{10N}$ is the 10-m, neutral-stability wind speed.
## 3. Results and Discussion
Using FluxEngine software, we produced monthly gridded global air-sea momentum
fluxes data. We calculated average momentum flux values separately for each month for the
global ocean, the NA Ocean, and its subsets: the Arctic sector of the NA and the West





Spitsbergen area (WS). Some of the parameterizations used were limited to a restricted wind
speed domain. We used them for all the global wind speed data to avoid data gaps for winds
that were too high or too low for a given parameterization (**Fig. 1**). However, circulation
models have the very same constraint and, therefore, the procedure we used emulated using
the parameterization in oceanographic and climate modeling.

Since wind velocity was used to estimate $C_D$, **Fig. 1** shows a wide range of empirical
formulas and **Fig. 6** shows annual mean wind speed $U_{10}$ (m s$^{-1}$) in the NA and the EA. The
differences between the oldest (eq. 8 - 10) and the newer (eq. 11, 13, 14) parameterizations
are distinct (**Fig. 1**). The $C_D$ values from the oldest parameterizations increased linearly with
wind speed since the results from newer ones are sinusoidal indicating decreases for winds in
the range of 0 - 10 m s$^{-1}$, after which they began increase. Under weak winds (< 10 m s$^{-1}$), the
drag coefficient values were significantly lower than under stronger winds (> 10 m s$^{-1}$), with
greater differences among all used parameterizations. At a wind value of about 15 m s$^{-1}$, the
results from eq. 9, 10, and 14 overlapped providing the same values for the drag coefficient
parameterizations. The annual mean wind speed in the NA is 10 m s$^{-1}$, and in the EA it is 8.5
m s$^{-1}$ (**Fig. 6**).

**Figure 2** presents maps of the mean boreal winter DJF and summer JJA momentum
fluxes for the chosen $C_D$ parameterizations (Wu, 1969 and A12 – the ones with the largest and
smallest $C_D$ values). The supplementary materials contain complete maps of annual and
seasonal means for all the parameterizations. The zones of the strongest winds are in the
extra-tropics in the winter hemisphere (southern for JJA and northern for DJF). The older Wu
(1969) parameterization produces higher wind stress values than A12 in both regions with
high and low winds and $C_D$ values are consistently higher for all wind speeds except the
lowest ones (which, after multiplying by U$^2$, produced negligible differences in wind stress
for the lowest winds). The average monthly values for each of the studied areas are shown in
**Fig. 3**. Generally, this illustrates that the newer the drag coefficient parameterization is, the
smaller the calculated momentum flux is. For global data (**Fig. 3a**), not much seasonal change
is noted, because the strongest winds are in fall and winter, but these seasons are the opposite
in the northern and southern hemispheres. The parameterization with the largest momentum
flux values for all months is that of Wu (1969), the oldest one, while the two
parameterizations with the lowest values are the newest ones (Large and Yeager, 2004 and
A12). For the NA (**Fig. 3b**), with is much more pronounced seasonal wind changes, the
situation is more complicated. With high winter winds, the A12 parameterization is no longer
the one that produces the smallest wind stress (it is actually in the middle of the seven).
However, for low summer winds, it is the lowermost outlier. Actually, in summer, the
constant $C_D$ value used by the NCEP/NCAR reanalysis produces the highest wind stress
values in the Na. The situation is similar for the EA (a subset of the NA), the wind stress
values of which are shown in **Fig. 3c**, and for the WS area (not show). In the Arctic summer,
A12 produces the least wind stresses, while all the other parameterizations look very similar
qualitatively (even more so in the Arctic than in the whole NA). Because the A12
parameterization behaves so distinctly differently with low winds, we also show seasonal
results for the tropical ocean (**Fig. 3d**). The seasonal changes are subdued for the whole



tropical ocean with the slight domination of the Southern Hemisphere (the strongest winds are
during the boreal summer) with generally lower momentum transfer values (monthly averages
in the range of 0.2 to 0.3 N m$^{-2}$ compared to 0.2 to 0.4 N m$^{-2}$ for the NA and 0.2 to 0.5 N m$^{-2}$
for the Arctic). The sequence of values for the parameterization is similar to that of the global
ocean, but there are differences. Here the NCEP/NCAR constant parameterization is the
second highest (instead of Wu, 1982 for the global ocean) while, unlike in the case of the
global ocean, A12 produces visibly lower values than does the Large and Yeager (2004)
parameterization.

We compared directly the results of the two parameterizations for the drag air-sea
relation that uses different dependencies (**Fig. 4**). For this estimation we chose the two most-
recent parameterizations (eq. 13 and 14) that showed the lowest values and change seasonally
depending on the area used. This comparison showed that the A12 parameterization
demonstrates almost zero sea surface drag for winds in the range of 3 - 5 m s$^{-1}$, which is
compensated for by a certain surplus value for strong winds. As a result, these months with
weak winds have significantly lower momentum flux values. This could be at statistical effect
of weak wind ocean areas having stable winds with waves traveling in the same direction as
the wind at similar speeds. The small drag coefficient values facilitate what Grachev and
Fairall (2001) describe as the transfer of momentum from the ocean to the atmosphere at wind
speeds of 2 - 4 m s$^{-1}$, which correspond to the negative drag coefficient value. Such events
require specific meteorologist conditions, but this strongly suggests that the average $C_D$ value
for similar wind speeds could be close to zero.

**Table 1** and **Fig. 5** present the average air-sea momentum flux values (in N m$^{-2}$) for all
the regions studied and all the parameterizations. All the values are also presented as
percentages of A12, which produced the lowest values for each region. A surprising result is
the proportionality of all the parameterizations for the global, the NA, and the Arctic regions
on annual scales (**Fig. 3** shows that this is not true on monthly scales). The spread of the
momentum flux results is 14 % in all three regions, and even flux values themselves are larger
in the NA than globally and larger in the Arctic than in the whole of the NA basin. The
smaller WS region, with winds that are, on average, weaker than those of the whole Arctic
(but stronger than those of the whole NA), had slightly different ratios of the resultant fluxes.
For the tropical ocean, which is included for comparison because of its weaker winds, the
spread in momentum flux values on an annual scale is 19 %. The spreads are even larger on
monthly scales (not shown). The difference between A12 and Wu (1969) and NCEP/NCAR
(the two parameterizations producing the largest fluxes on monthly scales) are 27 % and 29 %
for the NA (in July), 31 % and 36 % for the Arctic (in June), 42 % and 51 % for the WS
region (in July) and 23 % and 22 % for the tropical ocean (in April), respectively. Seasonality
in the tropics is weak, therefore, the smallest monthly difference of 16 % (July) is larger than
the difference for the global data in any month (the global differences between the
parameterizations have practically no seasonality). On the other hand, the smallest monthly
differences between the parameterizations in the NA, the Arctic, and the WS regions are all
7 %, in the month of the strongest winds (January).



Because the value of momentum flux is important for ocean circulation, its correct
calculation in coupled models is very important, especially in the Arctic, where cold halocline
stratification depends on the amount of mixing (Fer, 2009). We show that with the
parameterization used in modelling, such as the NCEP/NCAR constant parameterization and
Large and Yeager (2004), production stress results differ by about 5 %, on average (both in
the Arctic and globally), and the whole range of parameterizations leads to results that differ,
on average, by 14 % (more in low wind areas) and much more on monthly scales. One aspect
that needs more research is the fact that the newest parameterization, A12, produces less
momentum flux than all the previous ones, especially in lower winds (which, by the way,
continues the trend of decreasing values throughout the history of the formulas discussed).
The A12 parameterization is based on the largest set of  measurements of friction velocity as a
function of wind speed and utilizes the recently discovered fact that b in equation (8) is not
negligible. It also fits the observations that developed swell at low wind velocity has celerity
which leads to zero or even negative momentum transfer (Grachev and Fairall, 2001).
Therefore the significantly lower A12 results for the tropical ocean (the trade wind region)
and months of low winds elsewhere could mean that most momentum transfer calculations are
overestimated. This matter needs further study, preferably with new empirical datasets.

## 4. Conclusions

We show that the choice of drag coefficient parameterization can lead to significant
differences in resultant momentum flux (or wind stress) values. The differences between the
highest and lowest parameterizations are 14 % in the Arctic, the NA, and globally, and they
are higher in low winds areas. The parameterizations generally have a decreasing trend in the
resultant momentum flux values, with the most recent (Andreas et al., 2012) producing the
lowest wind stress values, especially at low winds, resulting in almost 20 % differences in the
tropics. The differences can be much larger on monthly scales, up to 29 % in the NA and 36 %
in the EA (in months of low winds) and even 50 % locally in the area west of Spitsbergen. For
months with the highest winds, the differences are smaller (about 7 % everywhere), but
because the flux values are largest with high winds this discrepancy is also important for air-
sea momentum flux values. Since momentum flux is an important parameter in ocean
circulation modeling, we believe more research is needed, and the parameterizations used in
the models possibly need upgrading.

## Acknowledgements

We would like to express our gratitude to Ed Andreas for inspiring us. His untimely departure
is an irreplaceable loss to the air-sea exchange community. We would also like to thank the
entire OceanFlux team. This publication was financed with funds from Leading National
Research Centre (KNOW) received by the Centre for Polar Studies for the period 2014–2018
and from OceanFlux Greenhouse Gases Evolution, a project funded by the European Space
Agency, ESRIN Contract No. 4000112091/14/I-LG.




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



**Table 1.** Average annual mean values of momentum flux (wind stress) [N m$^{-2}$] for all the studied regions and parameterizations. In each column the percentage values are normalized to A12, the parameterization that produced the smallest average flux values.

**Figure 1.** The drag coefficient parameterization used in the study (Eqs. 8-14) as a function of wind speed $U_{10}$ (m s$^{-1}$).

**Figure 2.** Maps of momentum flux [N m$^{-2}$] across the sea surface (wind stress) for boreal winters ((**a**) and (c)) and summers ((**b**) and (**d**)) for Wu (1969) and A12 drag coefficient parameterizations (the two parameterizations with the highest and lowest average values, respectively).

**Figure 3.** Monthly average momentum flux values [N m$^{-2}$] for (**a**) global ocean, (**b**) North Atlantic, (**c**) European Arctic, and (**d**) tropical ocean. The regions are defined in the text.

**Figure 4.** The drag coefficient values for Large and Yeager (2004) and Andreas et al., (2012) parameterization as a function of wind speed $U_{10}$ (m s$^{-1}$).

**Figure 5.** Annual average momentum flux values for (**a**) European Arctic and (b) Tropical ocean. The vertical solid line is the average of all seven parameterization and the dashed lines are standard deviations for the presented values. Global and the North Atlantic results are not shown because the relative values for different parameterizations are very similar (see Table 1), scaling almost identically between the basins.

**Figure 6.** Annual mean wind speed $U_{10}$ (m s$^{-1}$) in the study area—the North Atlantic and the European Arctic (north of the red line).

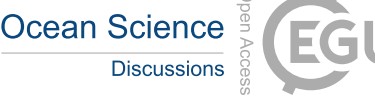



**Table 1.** Average annual mean values of momentum flux (wind stress) [N m$^{-2}$] for all the studied regions and parameterizations. In each column the percentage values are normalized to A12, the parameterization that produced the smallest average flux values.

|  | Global | North Atlantic | Arctic | W. Spitsbergen | Tropics |
|---|---|---|---|---|---|
| Wu (1969) | 0.322 (114 %) | 0.330 (114 %) | 0.375 (114 %) | 0.360 (114 %) | 0.261 (119 %) |
| Garratt (1977) | 0.307 (109 %) | 0.316 (109 %) | 0.358 (109 %) | 0.344 (110 %) | 0.251 (115 %) |
| Wu (1982) | 0.311 (110 %) | 0.320 (110 %) | 0.363 (110 %) | 0.349 (111 %) | 0.255 (117 %) |
| NCEP/NCAR | 0.303 (107 %) | 0.312 (107 %) | 0.353 (107 %) | 0.341 (108 %) | 0.258 (118 %) |
| Yelland & Taylor (1996) | 0.297 (105 %) | 0.306 (105 %) | 0.348 (106 %) | 0.335 (107 %) | 0.245 (112 %) |
| Large & Yeager (2004) | 0.285 (101 %) | 0.293 (101 %) | 0.333 (101 %) | 0.320 (102 %) | 0.236 (108 %) |
| Andreas et al., (2012) | 0.283 (100 %) | 0.290 (100 %) | 0.329 (100 %) | 0.314 (100 %) | 0.219 (100 %) |

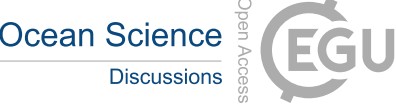



**Figure 1.** The drag coefficient parameterization used in the study (Eqs. 8-14) as a function of
wind speed $U_{10}$ (m s$^{-1}$).

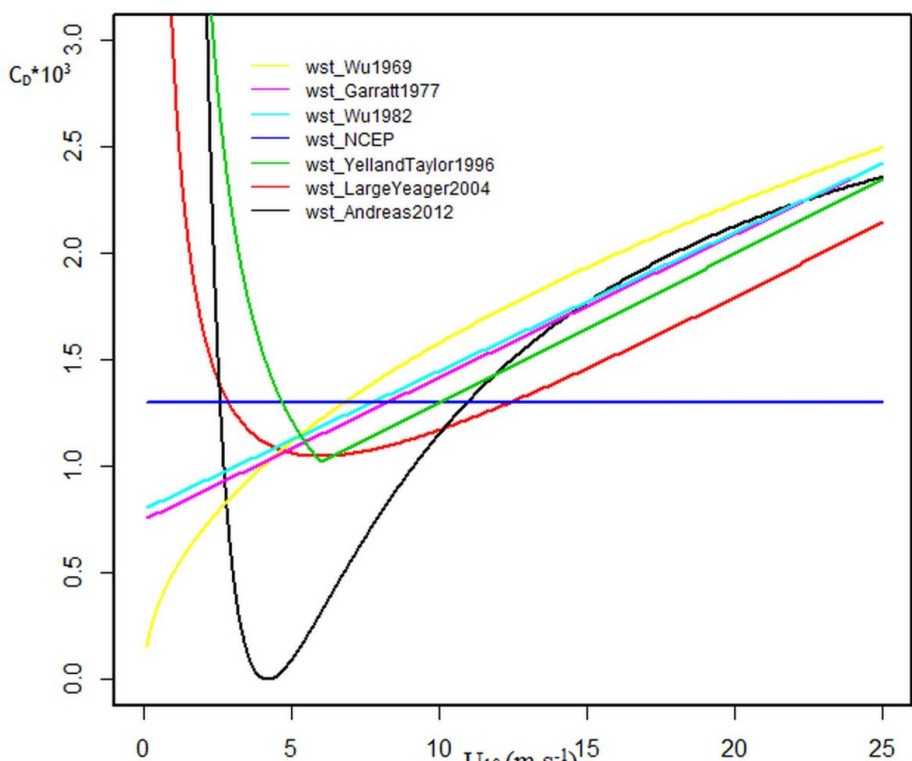













**Figure 2.** Maps of momentum flux [N m$^{-2}$] across the sea surface (wind stress) for boreal
winters ((**a**) and (**c**)) and summers ((b) and (**d**)) for Wu (1969) and A12 drag coefficient
parameterizations (the two parameterizations with the highest and lowest average values,
respectively).

(**a**) Wu, (1969)                           (**b**) Wu (1969)

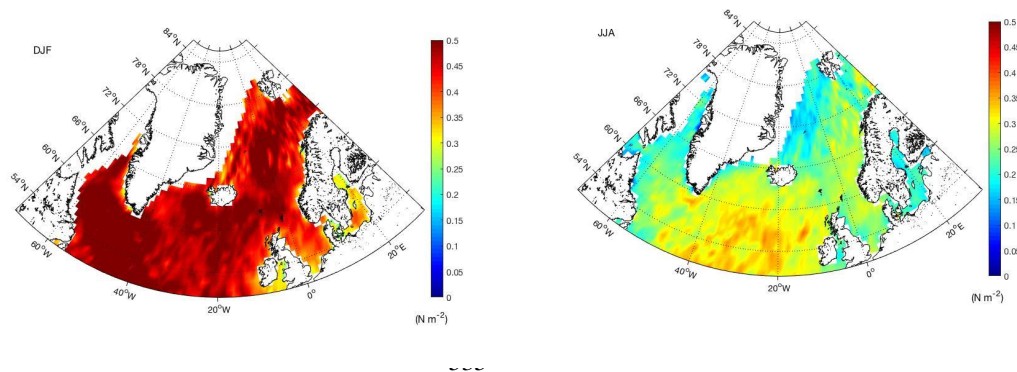

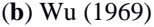

(**c**) Andreas, et al., (2012)                (**d**) Andreas, et al., (2012)

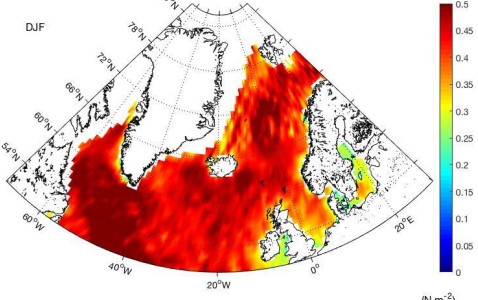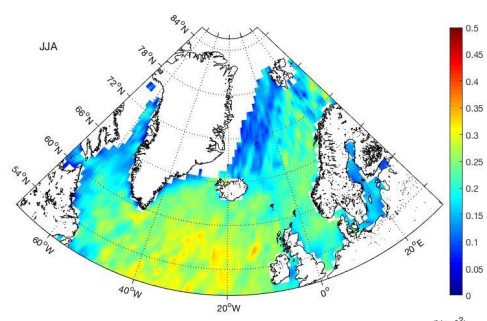





**Figure 3**. Monthly average momentum flux values [N m$^{-2}$] for (**a**) global ocean, (**b**) North Atlantic, (**c**) European Arctic, and (**d**) Tropical ocean. The regions are defined in the text.

(**a**)


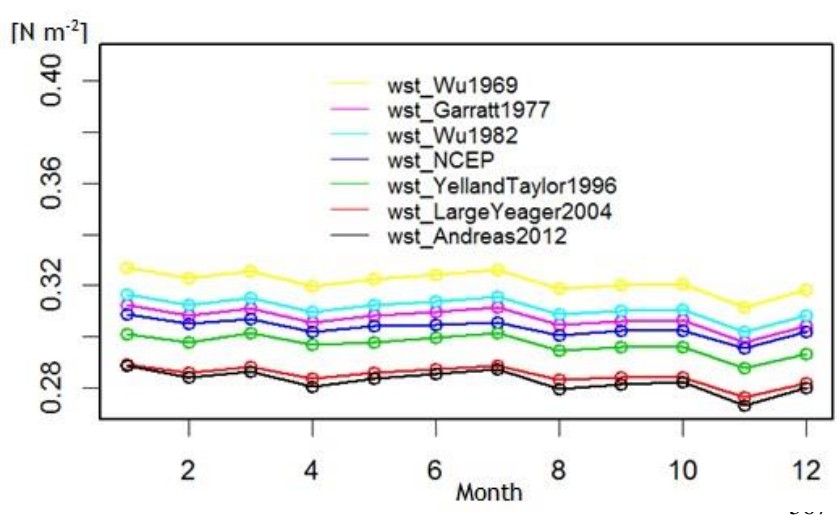

(**b**)

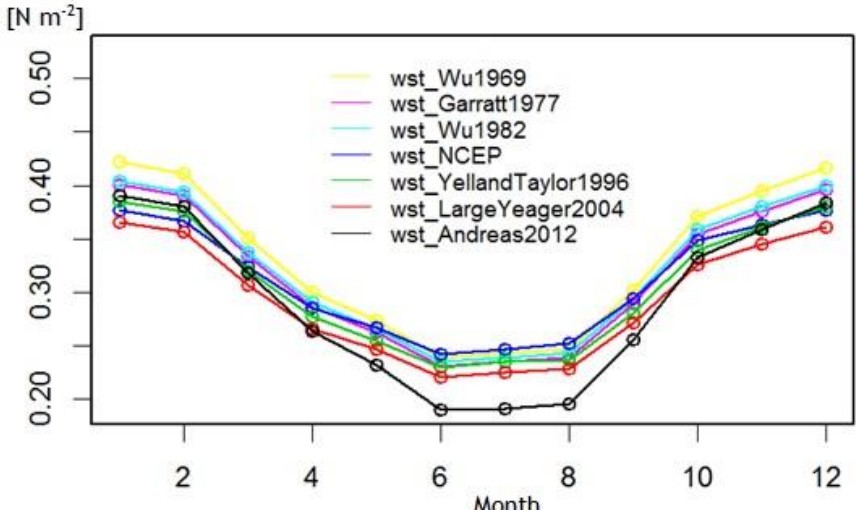





(c)

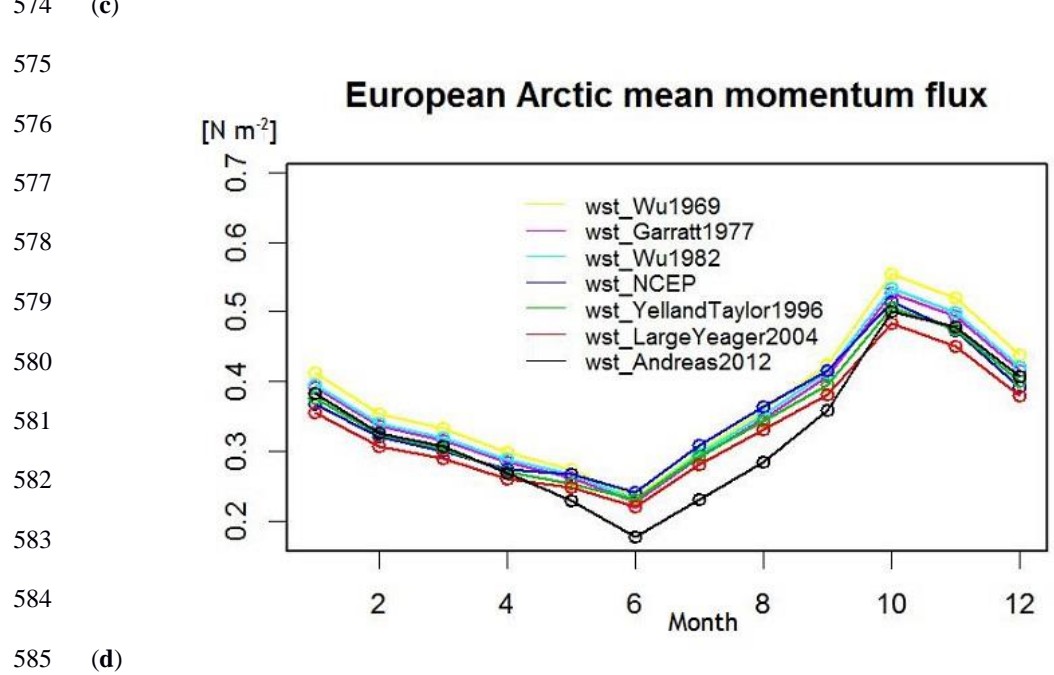

(d)



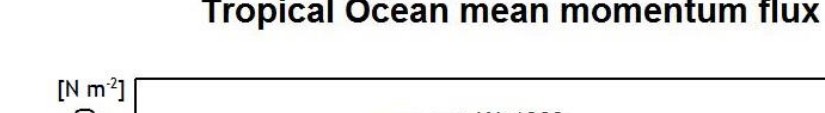
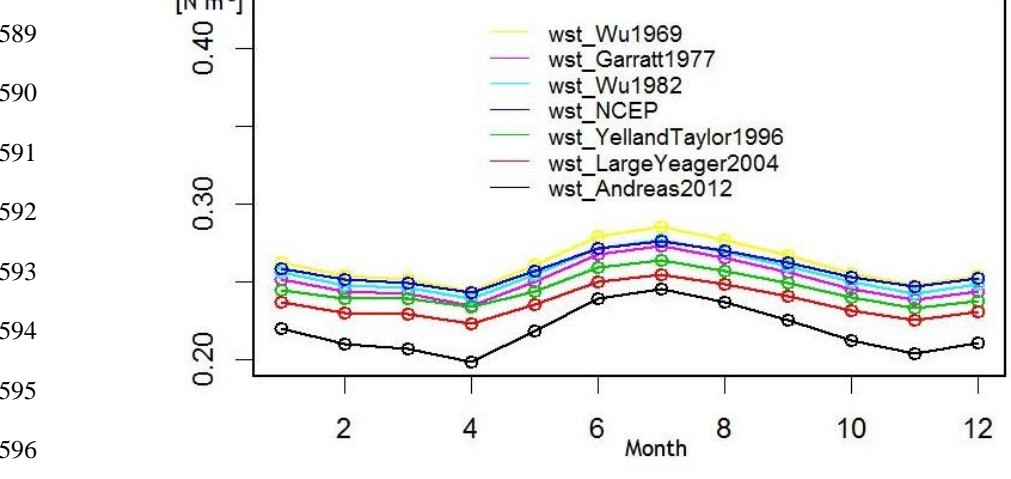






**Figure 4.** The drag coefficient values for Large and Yeager (2004) and Andreas et al., (2012)
parameterization as a function of wind speed $U_{10}$ (m s$^{-1}$).

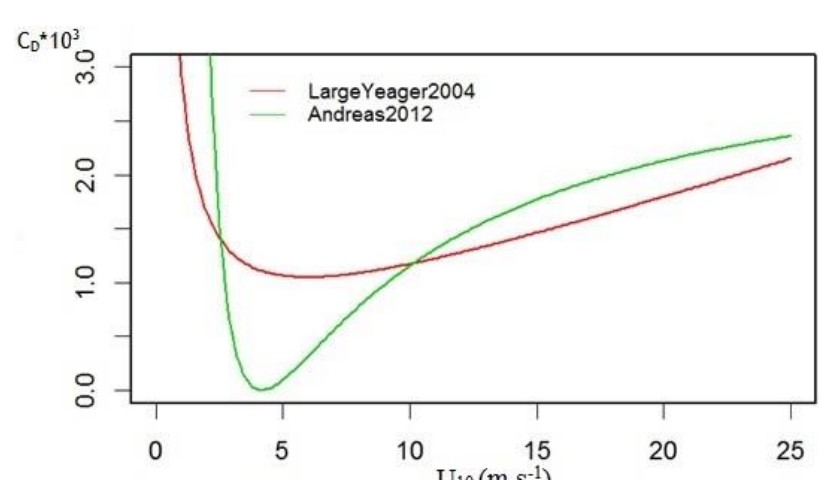





**Figure 5.** Annual average momentum flux values for (**a**) European Arctic and (**b**) Tropical
ocean. The vertical solid line is the average of all seven parameterization and the dashed lines
are standard deviations for the presented values. Global and the North Atlantic results are not
shown because the relative values for different parameterizations are very similar (see Table
1), scaling almost identically between the basins.
(**a**)

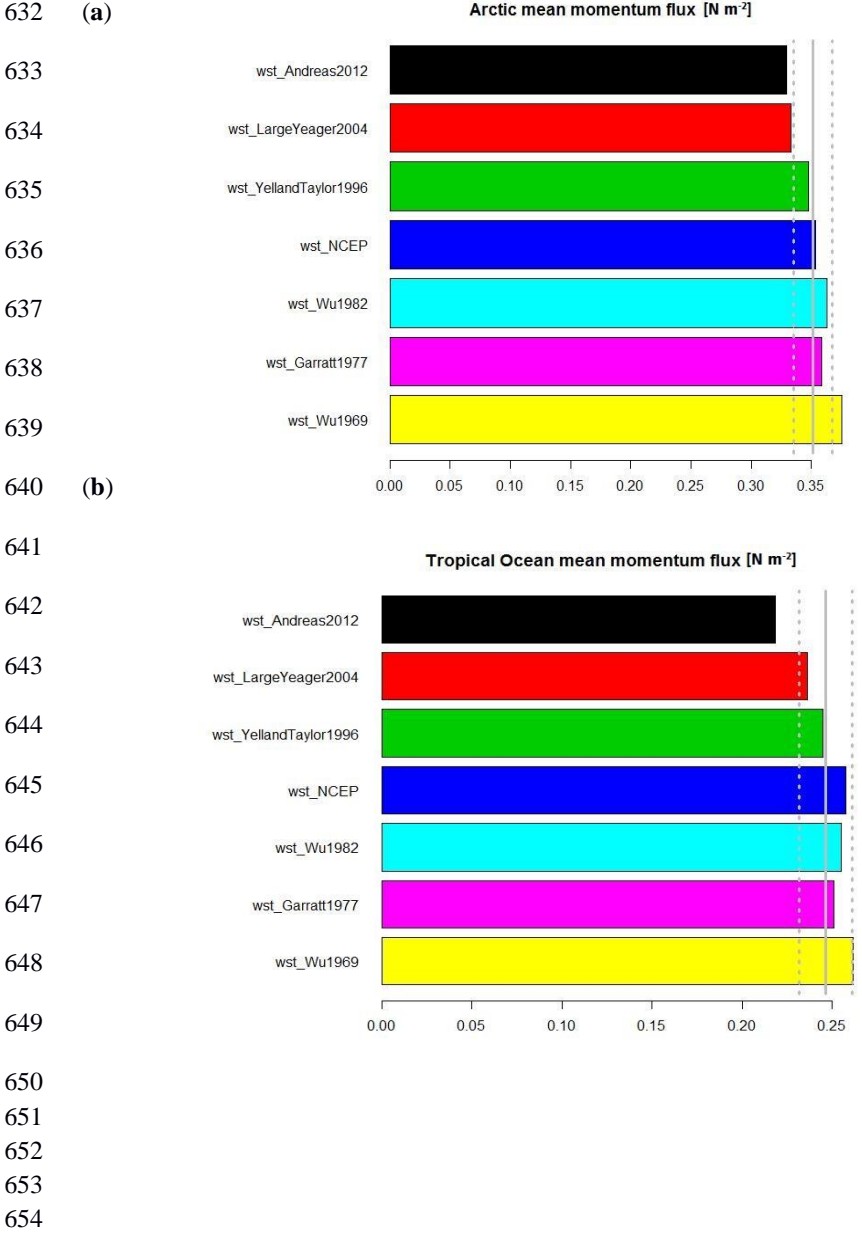

(**b**)



**Figure 6.** Annual mean wind speed $U_{10}$ (m s$^{-1}$) in the study area—the North Atlantic and the
European Arctic (north of the red line).

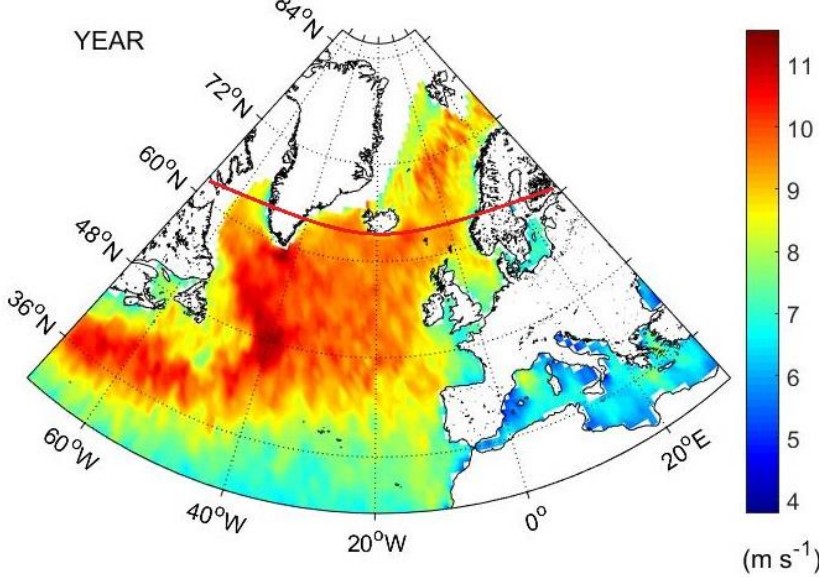