# Peer review of "Air-sea momentum flux climatologies: A review of drag relation for parameterization choice on wind stress in the North Atlantic and the European Arctic"

_Ocean Science, 2018_

## Referee Comment (RC1) · Anonymous Referee #1 · 19 Jul 2018

**Review** of the manuscript #os-2018-61

by Iwona Wrobel-Niedzwiecka et al.

"Air-sea momentum flux climatologies: A review of drag relation for parameterization choice on wind stress in the North Atlantic and the European Arctic"

Journal: Ocean Science

Summary: The authors use several parameterizations of the drag coefficient $C_D$ necessary to calculate wind stress (momentum flux) $\tau$ at the air-sea interface. The authors choose seven parameterizations of $C_D$ as a function of wind speed $U_{10}$. Processing tool (software) FluxEngine is used together with wind speed data from 1992 to 2010 to calculate $\tau$ in the North Atlantic (NA) and the European Arctic (EA). Area-averaged annual mean values of $\tau$ are tabulated for different $C_D(U_{10})$ parameterizations and compared to results obtained with the $C_D(U_{10})$ parameterization of Andreas et al. (2012, JPO).

Significance: The parameterization of $C_D$ has been a topic of enormous amount of research for many years. New studies try to add to the topic by reporting new data for $\tau$ and $C_D$; new parametrizations taking into account different processes and influences; comparisons between data and parametrizations; questioning the fundamental definition of $C_D$; determining $C_D$ under high wind speed conditions; explaining and/or modeling the leveling off under hurricanes etc. In this manuscript, the authors quantify the differences in $\tau$ values obtained with different $C_D(U_{10})$ parameterizations. In this sense, the topic of the manuscript is suitable for Ocean Science journal and useful for readers interested in this topic.

Evaluation: While each new research on $C_D$ and $\tau$ tries to add a bit more understating and propose improved $C_D(U_{10})$ expression(s), it is hard to add something truly new after decades of investigations. Thus, it becomes important to clearly state new developments and/or new insights when publishing on this topic. In this sense, this study does not add new knowledge. It is routine. Still, I see some usefulness of this manuscript in the tabulated $\tau$ differences for different $C_D(U_{10})$ as this can serve as a reference to readers regarding which of many available $C_D(U_{10})$ parameterizations to use. However, the manuscript needs much more work to be suitable for publication. In its present form, it lacks clear objective; could be better organized; and has weak conclusions. I recommend major revisions. Comments and suggestions follow.

Major comments:

1) The purpose/objective of the manuscript is not clearly formulated. The title suggests two things: a climatology and a review. Two more objectives are hinted in the text (details follow). Focusing on each of these possible objectives would require quite different analyses. With the purpose not well defined, none of the possible objectives is fully developed in the manuscript. Here are specifics.

a.  Is the objective a review of $C_D$ parameterizations, as the title suggests? If yes, then it is incomplete and without deep physical discussion of the progress and problems regarding parameterizing $C_D$. The authors state in Line 43 that it is a must to "take into account other physical processes," yet they then focus on "wind speed parameterizations, because wind speed is a parameter that is available in every atmospheric circulation model" (Lines 68-69). The authors initiate a review of formulas by dividing them in two groups (Lines 50-55), yet, again, stop short of further discussion on different formulation of roughness length $z_0$. I believe that review and physical discussion on $C_D$ are not the motivation of this work. After all, Andreas et al. (2012) and Edson et al. (2013, JPO, DOI: 10.1175/JPO-D-12-0173.1) provide comprehensive recent reviews of the status of parameterizing $C_D$.

b.  Is the objective a climatology of NA and EA, as the title suggests? If yes, this motivation is not well justified and there is no analysis of the results in climatological terms. If climatology is the objective, then the authors need to tell us why they focus on the NA and EA regions? What atmospheric and oceanic conditions does the $C_D$ parameterization need to represent well in these regions? If climatology is the goal, what is the temporal or spatial reference? It seems the chosen spatial references are the global ocean and the Tropics, to which the authors compare their results for NA and EA. But if $\tau$ is obtained globally and over many regions (Table 1), then why emphasize NA and EA in the title? Why look into differences due to $C_D$ formulation between northern regions and the Tropics, when it is certainly expected to have differences due to geography? As for a temporal reference, the authors should choose a period within or outside the 1992-2010 period which gives average atmospheric and oceanic conditions, not affected by long-term variations such as the North Atlantic Oscillation, which changes the position of jet stream and thus the wind and SST fields at the surface; these, in turn, change the wind stress. If climatology is the goal, then the authors should analyze their $\tau$ results for trends and variations over the 1992-2010 period. Should give annual as well as inter-annual variations. Finally, in my opinion, to provide a comprehensive regional climatology, the authors should analyze long-term $\tau$ values obtained with one, chosen $C_D$ formulation in order to clearly isolate climatologically-relevant variations.

c.  Is the objective to evaluate $C_D$ parameterizations and recommend a new one for use in circulation coupled models (Line 136)? A hint for such an objective comes from the authors' conclusion "the parameterizations used in the models possibly need upgrading" (Lines 332-333). If this is the objective, then the authors should give us a list of parameterizations used in different circulation models; discuss the advantages and limitations of these currently-used $C_D$ parameterizations; then demonstrate how other $C_D$ parameterizations would do better. For climate and circulation models, the $C_D$ parametrization is important for the mixing layer depth. So the authors should show how new $C_D$ parameterization would improve the modeling of the mixing layer. The manuscript offers limited information on what the current models use (Lines 136-140). There is no analysis on how $C_D$ would affect the performance of model variables related to $C_D$. So the conclusion in Lines 332-333 is not convincing for modelers.

d. Is the objective to demonstrate/justify the need of new measurements in NA and EA for improved $C_D$ parameterization in high latitudes? A hint for such an objective comes from Lines 188-189 regarding frequent ship deployment in EA, including "R/V Oceania, the ship of the institution the authors are affiliated with." If this is the objective, the manuscript would take completely different direction with discussion and analysis related to measuring methods and quality of data necessary for $C_D$. Of course, this is not the objective because the authors say in Lines 65-68 that their intention "is not to re-invent or formulate a new drag parameterization … but to revisit definition of the existing drag parameterization."

e. I am listing all these possible objectives only to make the point that the authors need a well stated objective in order to focus their analysis and discussion.

I believe the authors wish to assess $C_D$ parameterizations only to decide which one to use in some larger project. For such an assessment, the authors only need to clearly tell us why they consider the formulations (8)-(14) (i.e., no need of comprehensive review). They do not need climatology to make this assessment. One year data for $\tau$ is enough. However, to make the decision, the authors need not only to quantify $\tau$ differences (this is what the current results offer). They need also to make a thorough analysis what causes the differences. They need to evaluate how much of the differences come from: (i) different functional $C_D$ formulation; (ii) different quality of the data on which the parametrizations are based; and (iii) seasonal variations in NA and EA. The authors also need some reference to show them which $C_D$ formulation is suitable for NA and EA. Perhaps comparison of their results to regional data? Perhaps an investigation of how well a feature specific to NA and EA is represented when using different $C_D$ parameterizations? With such direction of the manuscript, the title may need revision to exclude claims on climatology and review.

2) There are several typos in the formulae that need fixing. Most importantly, it is necessary to check the coding for the calculations. These typos are as follows. In (2), $U_{10}^2$ is in the denominator. In (7), why $U_{10}$ is squared? The relationship between $u_*$ and $U_{10}$ is linear (Andresa et a., 2012, their eq. 1.10; Edson, et al., 2013, their eq. 22). In (14), need square on the wind ratio $(u_*/U_{10N})^2$; in the parenthesis, $U_{10N}^2$ is in the denominator; needs square on the parenthesis (compare to Andreas et a., 2012, their eq. 1.10).

3) Give better justification on choosing $C_D$ parameterizations (8)–(14). For example, it seems you have chosen $C_D$ parameterizations formulated as power law, linear, polynomial, constant. Why do you need (9) and (10)? They are so similar? Describe the merits of (11), (13) and (14), as well as their differences (e.g., data on which they are based). Do these formulations account implicitly for different processes in addition to $U_{10}$?

4) Suggest re-organizing the Introduction to include Lines 37-72 plus one paragraph on why you focus on NA and EA, then another paragraph clearly stating the objective of the study. Suggest combining Lines 73-154 with Lines 195-215 in one section dedicated on $C_D$ parameterizations. Only parts of the historical (incomplete) review in lines 73-154 are necessary. Start with the definitions in Lines 73-93. Then introduce

(8)–(14) one by one. Add information on MOST (Lines 115-122) and circulation models (Lines 136-140) only when they are needed, e.g., when you introduce (11) and (12), respectively. Finish the section with Lines 155-160. With this organization you will avoid the current inconsistency of presenting Fig. 1 with all parameterizations before they are described. Remove lines 122-127 and Lines 130-135 because you do not use Trenberth et al. (1989) and COARE algorithm. Unless you decide to use COARE 3.5 as a reference.

5) Section 3 "Result" is straightforward. It describes Table 1, maps, and seasonal graphs. To make these results useful, you need to extend the analysis of these data, discuss what causes the differences; and suggest which $C_D$ parameterizations is useful for NA and EA.

Additional comments:

Title: If possible (perhaps talk with the OS editor), revise the title to better reflect the purpose of your manuscript.

Abstract: Too long, dilutes what you did and what you have found. Suggest substantial shortening. Avoid giving references in the abstract. Refer to different $C_D$ parameterizations by their specific characteristics (e.g., power law, linear, etc), not by author.

Lines 18-19 and Lines 227-230: Oldest vs newest $C_D$ parameterization. This is not the most important difference. Frame your discussion around the functional form, the data they are based on, how well they represent low and high wind conditions.

Line 22-23: Suggest removing this sentence. This is common sense, no need to be in the abstract.

Line 30: "the sequence of values" is the least important thing to discuss about the differences. Discuss the physical behavior.

Line 76: Definition of $\tau$ is already given in Lines 42-43. Here, and many other places, remove repeated definitions.

Lines 89-90: Suggest removing this sentence, repeats definition given in Line 83.

Line 140: I guess you mean here equation (5), which assumes proportionality; (6) modifies (5) to linear relationship.

Line 144: I think you mean here equation (7), not (8). Eq. (7) needs correction (see Major comment 2).

Line 147: Fix symbol $U_{N10}$ to $U_{10N}$. Check all your math symbols for correctness and consistency.

Line 155: Fig 1 shows parameterizations whose equations are not yet introduced. Need to introduce (8)–(14) before referring to Fig. 1. See Major comment 4.

Line 168: Use symbol $U_{10}$ instead of re-defining it again.

Line 168-169: How these data on sea roughness are used? None of your equations (8)–(14) uses sea roughness. Why then introduce these data here?

Line 175: Use symbol $U_{10N}$ instead of re-defining it again.

Lines 177-179: You do not use wave data in (8)–(14), why do you introduce these data here?

Lines 180-183: Are all these details part of the FluxEngine software? Or are these done by you?

Lines 186, 226, 235: Suggest re-numbering Fig. 6 to Fig. 2, then all other figures. You refer to all other figures much later in the text.

Lines 195-215: Need to be introduced before Line 155 (see Major comment 4).

Line 217: "gridded global air-sea momentum" Why global when your emphasis is on NA and EA? Is global a good reference? You need representation of average conditions (either spatially or temporally averaged) for a reference. Need to work this out.

Line 229: Revise "sinusoidal". The decrease at low winds is not due to sinusoidal behavior.

Lines 246-248: Why looking into global values for seasonal variations when it is clear that opposite seasons cancel the variations? For seasonal variations, it is better to compare to Northern (or Southern) hemisphere.

Line 276: "could be at statistical effect" What do you mean? Suggest revision for clarity.

Line 286: What proportionality do you mean? Not clear.

Lines 329-331: Not clear what is your conclusion here. Please revise.

Line 333: "need upgrading" From what expression? To what expression? You make all these calculations but in the end you do not recommend what is good to use. See Major comment 1c.

Line 499: Average annual mean: Area averaged? Or over the time period 1992-2010? Not clear. Please revise here and in the text.

Figure 3a: Should show data for the Northern hemisphere if you want to use this as a reference for seasonal variations.

Fig. 4: Why do you need this figure? What more does it shown than Fig. 1?

Writing style and corrections:

Line 32: I guess you mean "Because A12 parameterization.."

Line 37: Suggest revision to read: "Wind stress at the air-sea interface influences the wind-wave interaction, including…"

Line 43: Suggest revising "must" to other word. "Must" is a firm request, which you do not follow in your subsequent considerations.

Lines 45-46: Suggest revising to read: "…fifty years, as the collection of flux data has increased, many empirical formulas…"

Line 61: Suggest revising "we chose to check" to "we investigate" or "we quantify"

Line 67: Suggest revising "accommodate" to "represent"

Lines 82-83: You have $u_*$ in italic and non-italic. Here, and everywhere, give mathematical symbols consistently.

Line 144: Abbreviation A12 should be introduced on first encounter, in line 50.

---

## Referee Comment (RC2) · Anonymous Referee #2 · 7 Aug 2018

The authors selectively review parameterizations of drag coefficient over the open ocean and evaluate the differences in stress that result. The methodology used is to take a satellite-derived data set of wind speed and use these as observations for bulk formulae evaluations of the momentum flux. Seven algorithms are compared. Differences in the momentum flux are (globally) about 14% and are comparable for different geographic regions, with some differences. This is a rather limited study. It uses scatterometer-derived neutral winds, which means the bulk flux algorithms can be directly compared without stability differences being a factor. This is fine, but it does

mean that the study does not yield much that is not already known or can be inferred from the equations themselves. Essentially the differences in momentum flux come straight out of the differences in the equations (illustrated in Fig. 1). The results are statistically-based, i.e. averages and mean differences, so there is not any link to atmosphere or ocean physics or dynamics. The choice of bulk flux algorithms to focus on is rather limited. There are a few well-known 'early' equations and then a couple of well-known later ones, but there are many very popular algorithms which are not evaluated, e.g. Smith (1988), the COARE algorithm (probably the most popular now), and others covered by other inter-comparisons. The study covers similar ground to that of Brunke et al. in a series of papers in the 2000s – see below for references. There is nothing wrong with this study, so I don't have any objections to it being published. However I am afraid I don't think it adds enough new to merit publication, so I cannot recommend it is published.

Minor comments There abstract is too long and there are numerous English errors.

References

Brunke, M. A., Fairall, C. W., Zeng, X., Eymard, L., & Curry, J. A. (2003). Which bulk aerodynamic algorithms are least problematic in computing ocean surface turbulent fluxes?. Journal of Climate, 16(4), 619-635.

Brunke, M. A., Zeng, X., & Anderson, S. (2002). Uncertainties in sea surface turbulent flux algorithms and data sets. Journal of Geophysical Research: Oceans, 107(C10), 5-1.

Brunke, M. A., Wang, Z., Zeng, X., Bosilovich, M., & Shie, C. L. (2011). An assessment of the uncertainties in ocean surface turbulent fluxes in 11 reanalysis, satellite-derived, and combined global datasets. Journal of Climate, 24(21), 5469-5493.

Smith, S. D. (1988). Coefficients for sea surface wind stress, heat flux, and wind profiles as a function of wind speed and temperature. Journal of Geophysical Research:

Oceans, 93(C12), 15467-15472.

---

## Author Comment (AC1) · 4 Oct 2018

Thank you for the reviews. We would ask you to reconsider our article for publication, because we have introduced a number of significant changes, following suggestion from reviewer no. 1, thanks to which the article is now better consulted, more understandable. You are right that Brunke et al., have done a lot of study in air-sea interaction, but their research are more extensive and concern a larger area. We have done our research by following among one of their conclusion, which is: Finally, a further investigation of the differences in the parameterization of the exchange coefficients in

the various algorithms would help in understanding some of the differences between the computed fluxes seen here. The aim of the manuscript is to evaluate how much the average monthly and annually momentum transfer values depend on the choice of CD parameterizations, in other words how the selected parameterization affects the total value of momentum fluxes for large reservoirs. This allows constraining the uncertainty caused by the parameterization choice. In order to achieve this, we used observed wind field for the regions of interest, namely the North Atlantic and the European Arctic, areas where European and Americans oceanographers, including us, operate. This is where most of studies that were basis of the parameterizations we use were performed. We did some comparisons to sub-tropical basins to see the difference in uncertainty caused by the formula choice between the main study regions and less studied subtopics. In our calculations, we do not clearly indicate which formula should be used in the future (impossible without new data) in the NA and the EU, but the simple fact that none of the parameterizations used now is final. We don't want to suggest end users any conclusions, because the differences in the parameterizations used are small, and our goal was to help them make an intelligent and deliberate decision about which parameterizations to use. We have chosen those 7 parameterizations as, in our opinion, they are the most commonly used in the literature during the last decade.

Please see the supplement material. We cointained there all the respond as well as corrected manuscript

Please also note the supplement to this comment:
https://www.ocean-sci-discuss.net/os-2018-61/os-2018-61-AC1-supplement.pdf

[Figure]

**Figure 1.** The drag coefficient parameterization used in the study (Eqs. 7-13) as a function of wind speed $U_{10}$ (m s$^{-1}$).

**Fig. 1.** Figure 1

[Figure]

1    **Figure 2.** Annual mean wind speed $U_{10}$ (m s$^{-1}$) in the study area—the North Atlantic and the
2    European Arctic (north of the red line).
3

[Figure]

**Fig. 2.** Figure 2

**Figure 3.** Maps of momentum flux [N m$^{-2}$] across the sea surface (wind stress) for boreal winters ((**a**) and (**c**)) and summers ((b) and (**d**)) for Wu (1969) and A12 drag coefficient parameterizations (the two parameterizations with the highest and lowest average values, respectively).

a) Wu, (1969)

[Figure]

b) Wu (1969)

**Fig. 3.** Figure 3

**Figure 4**. Monthly average momentum flux values [N m$^{-2}$] for (**a**) global ocean, (**b**) North Atlantic, (**c**) European Arctic, and (**d**) Tropical ocean. The regions are defined in the text.

a)

[Figure]

b)

[Figure]

**Fig. 4.** Figure 4

1   **Figure 5.** Area annual average momentum flux values for (**a**) European Arctic and (**b**)
2   Tropical ocean. The vertical solid line is the average of all seven parameterizations and the
3   dashed lines are standard deviations for the presented values. Global and the North Atlantic
4   results are not shown because the relative values for different parameterizations are very
5   similar (see Table 1), scaling almost identically between the basins.

6   a)

[Figure]

27   b)

**Fig. 5.** Figure 5

**Table 1.** Area average annual mean values of momentum flux (wind stress) [N m$^{-2}$] for all the studied regions and parameterizations. In each column the percentage values are normalized to A12, the parameterization that produced the smallest average flux values.

|  | Global | North Atlantic | Arctic | W. Spitsbergen | Tropics |
|---|---|---|---|---|---|
| Wu (1969) | 0.322 (114 %) | 0.330 (114 %) | 0.375 (114 %) | 0.360 (114 %) | 0.261 (119 %) |
| Garratt (1977) | 0.307 (109 %) | 0.316 (109 %) | 0.358 (109 %) | 0.344 (110 %) | 0.251 (115 %) |
| Wu (1982) | 0.311 (110 %) | 0.320 (110 %) | 0.363 (110 %) | 0.349 (111 %) | 0.255 (117 %) |
| NCEP/NCAR | 0.303 (107 %) | 0.312 (107 %) | 0.353 (107 %) | 0.341 (108 %) | 0.258 (118 %) |
| Yelland & Taylor (1996) | 0.297 (105 %) | 0.306 (105 %) | 0.348 (106 %) | 0.335 (107 %) | 0.245 (112 %) |
| Large & Yeager (2004) | 0.285 (101 %) | 0.293 (101 %) | 0.333 (101 %) | 0.320 (102 %) | 0.236 (108 %) |
| Andreas et al., (2012) | 0.283 (100 %) | 0.290 (100 %) | 0.329 (100 %) | 0.314 (100 %) | 0.219 (100 %) |

**Fig. 6.** Table 1

**Supplement:**

**RESPONSE TO REVIEWER 2**

October 4, 2018

The authors selectively review parameterizations of drag coefficient over the open ocean and evaluate the differences in stress that result. The methodology used is to take a satellite-derived data set of wind speed and use these as observations for bulk formulae evaluations of the momentum flux. Seven algorithms are compared. Differences in the momentum flux are (globally) about 14% and are comparable for different geographic regions, with some differences. This is a rather limited study. It uses scatterometer-derived neutral winds, which means the bulk flux algorithms can be directly compared without stability differences being a factor. This is fine, but it does mean that the study does not yield much that is not already known or can be inferred from the equations themselves. Essentially the differences in momentum flux come straight out of the differences in the equations (illustrated in Fig. 1). The results are statistically-based, i.e. averages and mean differences, so there is not any link to at-mosphere or ocean physics or dynamics. The choice of bulk flux algorithms to focus on is rather limited. There are a few well-known 'early' equations and then a couple of well-known later ones, but there are many very popular algorithms which are not evaluated, e.g. Smith (1988), the COARE algorithm (probably the most popular now), and others covered by other inter-comparisons. The study covers similar ground to that of Brunke et al. in a series of papers in the 2000s – see below for references. There is nothing wrong with this study, so I don't have any objections to it being published. However I am afraid I don't think it adds enough new to merit publication, so I cannot recommend it is published.

Minor comments There abstract is too long and there are numerous English errors.

**References**

Brunke, M. A., Fairall, C. W., Zeng, X., Eymard, L., & Curry, J. A. (2003). Which bulk aerodynamic algorithms are least problematic in computing ocean surface turbulent fluxes?. Journal of Climate, 16(4), 619-635.

Brunke, M. A., Zeng, X., & Anderson, S. (2002). Uncertainties in sea surface turbulent flux algorithms and data sets. Journal of Geophysical Research: Oceans, 107(C10), 5-1.

Brunke, M. A., Wang, Z., Zeng, X., Bosilovich, M., & Shie, C. L. (2011). An assessment of the uncertainties in ocean surface turbulent fluxes in 11 reanalysis, satellite-derived, and combined global datasets. Journal of Climate, 24(21), 5469-5493.

Smith, S. D. (1988). Coefficients for sea surface wind stress, heat flux, and wind pro- files as a function of wind speed and temperature. Journal of Geophysical Research: Oceans, 93(C12), 15467-15472.

Thank you for the reviews. We would ask you to reconsider our article for publication, because we have introduced a number of significant changes, following suggestion from reviewer no. 1, thanks to which the article is now better consulted, more understandable. You are right that Brunke et al., have done a lot of study in air-sea interaction, but their research are more extensive and concern a larger area. We have done our research by following among one of their conclusion, which is: *Finally, a further investigation of the differences in the parameterization of the exchange coefficients in the various algorithms would help in understanding some of the differences between the computed fluxes seen here.*

The aim of the manuscript is to evaluate how much the average monthly and annually momentum transfer values depend on the choice of CD parameterizations, in other words how the selected parameterization affects the total value of momentum fluxes for large reservoirs. This allows constraining the uncertainty caused by the parameterization choice. In order to achieve this, we used observed wind field for the regions of interest, namely the North Atlantic and the European Arctic, areas where European and Americans oceanographers, including us, operate. This is where most of studies that were basis of the parameterizations we use were performed. We did some comparisons to sub-tropical basins to see the difference in uncertainty caused by the formula choice between the main study regions and less studied subtopics. In our calculations, we do not clearly indicate which formula should be used in the future (impossible without new data) in the NA and the EU, but the simple fact that none of the parameterizations used now is final. We don't want to suggest end users any conclusions, because the differences in the parameterizations used are small, and our goal was to help them make an intelligent and deliberate decision about which parameterizations to use. We have chosen those 7 parameterizations as, in our opinion, they are the most commonly used in the literature during the last decade.

**Some of the major changes:**

We do our best to improve the manuscript, organize it better than it was, clearly state the objective and conclusion, also mark what new it adds to our knowledge. As the original title could cause confusion and did not clearly define the paper purpose. Therefore we change it in the revised version. The new title now is: Effect of drag coefficient formula choice on wind stress climatology in the North Atlantic and the European Arctic The reviewer no 1. pointed out that the purpose of the manuscript was not clearly formulated, so we corrected it and added properly information and corrections inside the text:

L23-32 Confirming and explaining the nature and consequence of the interaction between the atmosphere and ocean is one of the great challenge in climate and sea research. These two sphere are coupled which lead to variations covering time scales from minutes to even millennia. The purpose of this article is to examine, using a modern set of software processing tools called the FluxEngine, the nature of the fluxes of momentum across the sea surface over the North Atlantic and the European Arctic. These fluxes are important to determine of current system and sea state conditions. Our goal is to evaluate how much the average monthly and annually momentum transfer values depend on the choice of  $C_D$ , using the actual wind field from the North Atlantic and the European Arctic, and demonstrate existing differences as a result of the formula used.

L134-145 In this paper we investigate how the relevant or most commonly used parameterizations for drag coefficient ( $C_D$ ) affect to value of momentum transfer values, especially in the North Atlantic (NA) and the European Arctic (EA). Our task was to demonstrate existing differences as a result of the formula used how big they can be. As is widely known, the exact equation that describes the connection between the drag coefficient and wind speed depends on the author (Geernaert, 1990). Our intention here is not to re-invent or formulate a new drag parameterization for the NA or the EA, but to revisit the existing definition of drag parameterization, and, using satellite data, to investigate how existing formulas represent the environment in the North. We concentrated on wind speed parameterizations, because wind speed is a parameter that is available in every atmospheric circulation model. Therefore, it is used in all air-sea flux parameterizations, and presently it is used even when sea state provides a closer physical coupling to the drag coefficient (for review see Geernaert et al., 1986).

**Effect of drag coefficient formula choice on wind stress climatology in the North Atlantic and the European Arctic**

Iwona Wrobel-Niedzwiecka, Violetta Drozdowska, Jacek Piskozub1

1 Institute of Oceanology, Polish Academy of Science, ul. Powstancow Warszawy 55, 81-712 Sopot, Poland corresponding author: iwrobel@iopan.gda.pl

Key points: drag coefficient, European Arctic, North Atlantic, parameterizations

**3 Abstract**

4 In this paper we have chosen to check the differences between the relevant or most commonly 5 used parameterizations for drag coefficient  $(C_D)$  for the momentum transfer values, especially in the North Atlantic (NA) and the European Arctic (EA). We studied monthly values of air-6 7 sea momentum flux resulting from the choice of different drag coefficient parameterizations, 8 adapted them to momentum flux (wind stress) calculations using SAR wind fields, sea-ice 9 masks, as well as integrating procedures. We compared the resulting spreads in momentum flux to global values and values in the tropics, an area of prevailing low winds. We show that the 10 choice of drag coefficient parameterization can lead to significant differences in resultant 11 momentum flux (or wind stress) values. We found that the spread of results stemming from the 12 choice of drag coefficient parameterization was 14 % in the Arctic, the North Atlantic and 13 14 globally, but it was higher (19%) in the tropics. On monthly time scales, the differences were larger at up to 29 % in the North Atlantic and 36 % in the European Arctic (in months of low 15 16 winds) and even 50 % locally (the area west of Spitsbergen). When we choose the 17 parameterizations which increased linearly with wind speed (7-9) momentum flux were largest for all months, in compare to values from the two parameterizations which increase with wind 18 speed sinusoidal (12 and 13), in both regions with high and low winds and CD values were 19 consistently higher for all wind speeds. As the one of power law parameterization (13) behaves 20 21 so distinctly differently with low winds, we showed seasonal results for the tropical ocean, which were subdued for the whole region, with monthly averages in the range of 0.2 to 0.3 N 22  $m^2$ . 23

**24 **1. Introduction**

Confirming and explaining the nature and consequence of the interaction between the 25 atmosphere and ocean is one of the great challenge in climate and sea research. These two 26 sphere are coupled which lead to variations covering time scales from minutes to even 27 millennia. The purpose of this article is to examine, using a modern set of software processing 28 tools called the FluxEngine, the nature of the fluxes of momentum across the sea surface over 29 the North Atlantic and the European Arctic. These fluxes are important to determine of current 30 system and sea state conditions. Our goal is to evaluate how much the average monthly and 31 annually momentum transfer values depend on the choice of CD, using the actual wind field 32 from the North Atlantic and the European Arctic, and demonstrate existing differences as a 33 result of the formula used. 34

35 The ocean surface mixed layer is a region where kinematic forcing affects the exchange of horizontal momentum and controls transport from the surface to depths (Gerbi et al., 2008, 36 Bigdeli et al., 2017). Any attempt to properly model the momentum flux from one fluid to 37 another as the drag force per unit area at the sea surface (surface shear stress,  $\tau$ ) take into account 38 other physical processes responsible for generating turbulence such as boundary stress, 39 40 boundary buoyancy flux, and wave breaking (Rieder et al., 1994, Jones and Toba, 2001). Fluxes across the sea surface usually depend nonlinearly on the relevant atmospheric or oceanic 41 parameters. Over the past fifty years, as the collection of flux data has increased, many 42 empirical formulas have been developed to express the ocean surface momentum flux as a 43 44 relationship between non-dimensional drag coefficient ( $C_D$ ), wind speed ( $U_{10}$ ), and surface roughness  $(z_0)$  (Wu 1969, 1982; Bunker, 1976; Garratt, 1977; Large and Pond, 1981; Trenberth 45 et al., 1989; Yelland and Taylor, 1996, Donelan et al., 1997; Kukulka et al., 2007; Andreas et 46 al., 2012). These formulas can be divided into two groups. One group of theories gives the  $C_D$ 47 48 at level z in terms of wind speed and possibly one or more sea-state parameters (for example, 49 Geernaert et al., 1987, Yelland and Taylor, 1996, Enriquez and Friehe, 1997), while the second group provides formulas for roughness length  $z_0$  in terms of atmospheric and sea-state 50 parameters (for example, Wu, 1969, Donelan et al., 1997, Andreas et al., 2012 (further referred 51 52 to as A12)).

As the exchange of air-sea momentum is difficult to measure directly over the ocean
meteorologist and oceanographers often rely on bulk formulas parameterized by Taylor (1916),
that relate the fluxes to averaged wind speed through transfer coefficients:

$$\tau = \rho C_{Dz} U_z^2 \tag{1}$$

57 where  $\tau$  is the momentum flux of surface stress,  $\rho$  is air density,  $C_{Dz}$  is the non-dimensional 58 drag coefficient appropriate for *z* height, and Uz is the average wind speed at some reference 59 height *z* above the sea.  $C_{Dz}$  is commonly parameterized as a function of mean wind speed (m s-1) for neutral-stability at a 10 m reference height above mean sea level (Jones and Toba, 2001), 60 1) for neutral-stability at a 10 m reference height above mean sea level (Jones and Toba, 2001), 61 which is identified as  $C_{DN10}$  or  $C_{D10}$  (this permits avoiding deviation for the vertical flow from 62 the logarithmic law):

63
$$C_{DN10} = \frac{\tau}{\rho U_{10}^2} = \left(\frac{u_*}{U_{10}}\right)^2$$
(2)

64 where  $u_*$  is friction velocity. Alternatively, the neutrally stratified momentum flux can be 65 determined from the logarithmic profile, thus Eq. 1 can be express as:

66
$$C_{DN10} = [\kappa/\ln(10/z_0)]^2$$
 (3)

67 where  $z_0$  (m) is the aerodynamic roughness length, which is the height, above the surface to 68 define the measure of drag at which wind speed extrapolates to 0 on the logarithmic wind profile 69 (Andreas et al., 2012), and  $\kappa$  is von Kármán constant ( $\kappa$ =0.4).

70 At the same time, we can define the value of friction velocity by the following equation:

$$\tau = \rho \, u_*^2 \tag{4}$$

72 Comparison with bulk formula (1) leads to the equation:

73
$$u_*^2 = C_{D10} U_{10}^2$$
 (5)

Some of the first studies (Wu, 1969, 1982, Garrat, 1977) focused on the relationship between wind stress and sea surface roughness, as proposed by Charnock (1955), and they formulated (for winds below 15 m s-1) the logarithmic dependence of the stress coefficient on wind velocity (measured at a certain height) and the von Kármán constant. Currently common parameterizations of the drag coefficient are a linear function of 10 m wind speed (U10), and the parameters in the equation are determined empirically by fitting observational data to a curve. The general form is expressed as (Guan and Xie, 2004):

$$C_D 10^3 = (a + bU_{10}) \tag{6}$$

82 In this work our focus is on the fluxes of average values using seven different drag 83 coefficient parameterizations ( $C_D$ ), chosen for their importance for the history of the field out 84 of many published within the last half century (Bryant and Akbar, 2016).

85
$$10^3 \cdot C_{D10} = 0.5U_{10}^{0.5}$$
 for 1 m s-1 <  $U_{10}$  < 15 m s-1 (7)
(Wu, 1969)

87
$$10^3 \cdot C_{DN10} = 0.75 + 0.067 U_{10}$$
 for 4 m s-1 < U < 21 m s-1 (8)
88 (Garratt,

89 1977)

81

90
$$10^3 \cdot C_{D10} = (0.8 + 0.065U_{10})$$
 for  $U_{10} > 1 \text{ m s}^{-1}$  (9)
91 (Wu, 1982)

92
$$10^3 \cdot C_{DN10} = 0.29 + \frac{3.1}{U_{10N}} + \frac{7.7}{U_{10N}^2}$$
 for 3 m s-1 <  $U_{10N} < 6$  m s-1 (10)
93 for  $3 = 0.60 + 0.070 U$  for  $6 = 0.71 + 0.070 U$  for  $6 = 0.71 + 0.070 U$

93
$$10^{\circ} \cdot c_{DN10} = 0.60 + 0.070 U_{10N}$$
 for 6 m s <  $U_{10N} < 26$  m s
94 (Yelland and Taylor, 1996)

95
$$10^3 \cdot C_D = 1.3$$
 everywhere (11)
96 (NCEP/NCAR)

97
$$10^3 \cdot C_{DN10} = \frac{2.7}{U_{10N}} + 0.142 + 0.076U_{10N}$$
 everywhere
98 (L

(Large and Yeager, 2004)

(12)

99
$$C_{DN10} = (\frac{u^*}{U_{10N}})^2 = a^2 (1 + \frac{b}{a} U_{10N})^2$$
 everywhere (13)
100 where  $a = 0.0583, b = -0.243$  (Andreas et al., 2012)

where  $C_{DN10}$  is the expression of neutral-stability (10-m drag coefficient),  $C_{D10}$  is the drag 101 coefficient dependent on surface roughness,  $U_{10}$  is the mean wind speed measured at 10 m above 102 the mean sea surface, U10N is the 10-m, neutral-stability wind speed. All of them are generated 103 from the vertical wind profile, but they differ in the formulas used. Two of the parameterization 104 105 which we chosen are formulated as power-law of the relationship between  $C_D$  and  $U_{10}$  (eq. 7) and 13), three are formulated as linear-law (eq. 8, 9 10 for light winds, and 12), and one as 106 107 constant value of the relationship (eq. 11). All the above studies propose different parameterizations (see Fig. 1) of the drag coefficient and the function of wind speed, which 108

reflects the difficulties in simultaneously measuring at high sea stress (or friction velocity) andwind speed.

[revised manuscript text omitted]

**3. Results and Discussion**

Using the FluxEngine software, we produced global gridded monthly air-sea 179 momentum fluxes and from these we have extracted the values for the study region, the global 180 ocean, the NA Ocean, and its subsets: the Arctic sector of the NA and the West Spitsbergen 181 182 area (WS). Some of the parameterizations used were limited to a restricted wind speed domain. 183 We used them for all the global wind speed data to avoid data gaps for winds that were too high 184 or too low for a given parameterization (Fig. 1). However, circulation models have the very same constraint and, therefore, the procedure we used emulated using the parameterization in 185 186 oceanographic and climate modeling.

187 Since wind velocity was used to estimate  $C_D$ , **Fig. 1** shows a wide range of empirical 188 formulas and **Fig. 2** shows annual mean wind speed  $U_{10}$  (m s-1) in the NA and the EA. The 189 differences between the parameterizations are distinct (**Fig. 1**). The  $C_D$  values from the

parameterizations 7 - 9 increased linearly with wind speed since the results from the 190 parameterizations 10,12,13 are characterized by sinusoidal distribution and indicating 191 decreases for winds in the range of 0 - 10 m s-1, after which they began increase. Despite many 192 measurements, the drag coefficient still has wide variability at low and moderate wind speeds. 193 Our research has showed that al lower wind values ( $<10 \text{ m s}^{-1}$ ) the differences between the drag 194 coefficient parameterizations are greater than at higher speeds (> 10 m s-1) and the most outlier 195 results are those obtained from the power law parameterization of Andreas et al., (2012). The 196 197 lower the wind speed, the higher uncertainty are, and at low winds it is uncertainty by a factor of 0.5-1.5 depending on the formula used, while at moderate winds it is uncertainty by a factor 198 of 1.5-2.0 (Fig. 1). At a wind value of about 15 m s-1, the results from eq. 8, 9, and 13 overlapped 199 providing the same values for the drag coefficient parameterizations. Additionally, we 200 compared directly the results of the two parameterizations for the drag air-sea relation that uses 201 202 different dependencies (Fig. 1). For this estimation we chose the two most-recent parameterizations (eq. 12 and 13) that showed the lowest values and change seasonally 203 depending on the area used. As a result, these months with weak winds have significantly lower 204 205 momentum flux values, which could be the effect of statistically weaker wind in ocean areas having stable winds with waves traveling in the same direction as the wind at similar speeds. 206 207 Comparison showed that the A12 parameterization demonstrates almost zero sea surface drag for winds in the range of 3 - 5 m s-1, which is compensated for by a certain surplus value for 208 strong winds. The small drag coefficient values facilitate what Grachev and Fairall (2001) 209 describe as the transfer of momentum from the ocean to the atmosphere at wind speeds of 2 - 4 210 m s-1, which correspond to the negative drag coefficient value. Such events require specific 211 meteorologist conditions, but this strongly suggests that the average  $C_D$  value for similar wind 212 speeds could be close to zero. The annual mean wind speed in the NA is 10 m s-1, and in the 213 EA it is  $8.5 \text{ m s}^{-1}$  (Fig. 2). 214

Figure 3 presents maps of the mean boreal winter DJF and summer JJA momentum 215 fluxes for the chosen  $C_D$  parameterizations (Wu, 1969 and A12 – the ones with the largest and 216 smallest  $C_D$  values). The supplementary materials contain complete maps of annual and 217 seasonal means for all the parameterizations. The zones of the strongest winds are in the extra-218 219 tropics in the winter hemisphere (southern for JJA and northern for DJF). The older Wu (1969) 220 parameterization produces higher wind stress values than A12 in both regions with high and low winds and  $C_D$  values are consistently higher for all wind speeds except the lowest ones 221 (which, after multiplying by  $U^2$ , produced negligible differences in wind stress for the lowest 222 winds). The average monthly values for each of the studied areas are shown in Fig. 4. Generally, 223 this illustrates that the sinusoidal the drag coefficient parameterization is, the smaller the 224 225 calculated momentum flux is. For global data (Fig. 4a), not much seasonal change is noted, 226 because the strongest winds are in fall and winter, but these seasons are the opposite in the northern and southern hemispheres. The parameterization with the largest momentum flux 227 values for all months is that of Wu (1969), the linear one, while the two parameterizations with 228 the lowest values are the sinusoidal ones (Large and Yeager, 2004 and A12). For the NA (Fig. 229 4b), with is much more pronounced seasonal wind changes, the situation is more complicated. 230 With high winter winds, the A12 parameterization is no longer the one that produces the 231 smallest wind stress (it is actually in the middle of the seven). However, for low summer winds, 232

it is the lowermost outlier. Actually, in summer, the constant  $C_D$  value used by the NCEP/NCAR 233 reanalysis produces the highest wind stress values in the NA. The situation is similar for the 234 235 EA (a subset of the NA), the wind stress values of which are shown in Fig. 4c, and for the WS 236 area (not show). In the Arctic summer, A12 produces the least wind stresses, while all the other parameterizations look very similar qualitatively (even more so in the Arctic than in the whole 237 NA). Because the A12 parameterization behaves so distinctly differently with low winds, we 238 239 also show seasonal results for the tropical ocean (Fig. 4d). The seasonal changes are subdued 240 for the whole tropical ocean with the slight domination of the Southern Hemisphere (the strongest winds are during the boreal summer) with generally lower momentum transfer values 241 (monthly averages in the range of 0.2 to 0.3 N m-2 compared to 0.2 to 0.4 N m-2 for the NA and 242 0.2 to 0.5 N m-2 for the Arctic). The sequence of values for the parameterization is similar to 243 that of the global ocean, but there are differences. Here the NCEP/NCAR constant 244 245 parameterization is the second highest (instead of Wu, 1982 for the global ocean) while, unlike in the case of the global ocean, A12 produces visibly lower values than does the Large and 246 247 Yeager (2004) parameterization.

248 Table 1 and Fig. 5 present the annual average air-sea momentum flux values (in N m 249  $^{2}$ ) for all the all regions studied and all the parameterizations. The results show that the annual 250 North Atlantic momentum fluxes, depending on the formula used, varies from -0.0.290 N m-2 for A12 to 0.333 N m-2 for Wu (1969). In the case of global annual average, the values are -251 0.283 and 0.322, respectively. Table 1 shows also the same data "normalized" to the A12 data 252 253 (presented as percentages of A12, which produced the lowest values for each region), which 254 allows us to visualize the relative differences. A surprising result is the annual ratios of the 255 parameterizations values for the global, the NA, and the Arctic regions (Fig. 4 shows that this is not true on monthly scales). The spread of the momentum flux results is 14 % in all three 256 257 regions, and even flux values themselves are larger in the NA than globally and larger in the Arctic than in the whole of the NA basin. In the NA region with winds stronger than average 258 for world ocean, the formula giving highest momentum transfer results are the ones with highest 259 values for strong winds, with exception of Andreas et al. (2012) which is lower due to its low 260 261 values for lower winds speeds. The smaller WS region, with winds that are, on average, weaker 262 than those of the whole Arctic (but stronger than those of the whole NA), had slightly different ratios of the resultant fluxes. For the tropical ocean, which is included for comparison because 263 of its weaker winds, the spread in momentum flux values on an annual scale is 19 %. The 264 spreads are even larger on monthly scales (not shown). The difference between A12 and Wu 265 266 (1969) and NCEP/NCAR (the two parameterizations producing the largest fluxes on monthly scales) are 27 % and 29 % for the NA (in July), 31 % and 36 % for the Arctic (in June), 42 % 267 and 51 % for the WS region (in July) and 23 % and 22 % for the tropical ocean (in April), 268 respectively. Seasonality in the tropics is weak, therefore, the smallest monthly difference of 269 270 16 % (July) is larger than the difference for the global data in any month (the global differences between the parameterizations have practically no seasonality). On the other hand, the smallest 271 monthly differences between the parameterizations in the NA, the Arctic, and the WS regions 272 are all 7 %, in the month of the strongest winds (January). 273

274 Because the value of momentum flux is important for ocean circulation, its correct calculation in coupled models is very important, especially in the Arctic, where cold halocline 275 276 stratification depends on the amount of mixing (Fer, 2009). We show that with the 277 parameterization used in modelling, such as the NCEP/NCAR constant parameterization and Large and Yeager (2004), production stress results differ by about 5 %, on average (both in the 278 279 Arctic and globally), and the whole range of parameterizations leads to results that differ, on 280 average, by 14 % (more in low wind areas) and much more on monthly scales. One aspect that needs more research is the fact that the newest parameterization, A12, produces less momentum 281 flux than all the previous ones, especially in lower winds (which, by the way, continues the 282 trend of decreasing values throughout the history of the formulas discussed). The A12 283 284 parameterization is based on the largest set of measurements of friction velocity as a function 285 of wind speed and utilizes the recently discovered fact that b in equation (7) is not negligible. It also fits the observations that developed swell at low wind velocity has celerity which leads 286 287 to zero or even negative momentum transfer (Grachev and Fairall, 2001). Therefore, the 288 significantly lower A12 results for the tropical ocean (the trade wind region) and months of low 289 winds elsewhere could mean that most momentum transfer calculations are overestimated. This 290 matter needs further study, preferably with new empirical datasets.

**291 **4.** Conclusions**

292 In the present work the evaluation of how the selected parameterization affects the total 293 value of momentum fluxes for large reservoirs was assessed. This allows constraining the 294 uncertainty caused by the parameterization choice. In order to achieve this we calculated 295 monthly and annually average momentum fluxes using a set of software processing tools called 296 the FluxEngine in the North Atlantic (NA) and the European Artic (EA). The NA was defined 297 as all sea surface areas in the Atlantic sector north of 30° N, and the EA was sea areas north of 64° N. Based on our results, we still do not know which one of the parameterizations can be 298 299 reccomend as the most suitable for the NA and the EU study. Further investigation of the 300 differences in the parameterization of the exchange coefficient in the various algorithm would help in resolving this problem. 301

Bespite many measurements, the drag coefficient still has wide variability at low and moderate wind speeds. The lower the wind speed, the higher uncertainty are, and at low winds it is uncertainty by a factor of 0.5-1.5 depending on the formula used, while at moderate winds it is uncertainty by a factor of 1.5-2.0 (Fig. 1). The annual mean wind speed in the NA is 10 m  $s^{-1}$ , and in the EU it is 8.5 m s-1.

307 We show that the choice of drag coefficient parameterization can lead to significant differences in resultant momentum flux (or wind stress) values. Comparing the values of 308 momentum flux across the sea surface from the power law parameterization, it showed that in 309 310 both regions, with low and high winds, the parameterizations specified for all winds speeds (eq. 311 13) has lower values of wind stress than the parameterizations specified for light winds (eq. 7). 312 In the Arctic, the NA, and globally the differences between the wind stress, depend on formula 313 used, are 14 % and they are higher in low winds areas. The parameterizations generally have a decreasing trend in the resultant momentum flux values, with the most recent (Andreas et al., 314 315 2012) producing the lowest wind stress values, especially at low winds, resulting in almost 20

% differences in the tropics (Table1). The differences can be much larger on monthly scales, 316 up to 29 % in the NA and 36 % in the EA (in months of low winds) and even 50 % locally in 317 the area west of Spitsbergen. For months that have the highest average winds, the percentage 318 differences are smaller (about 7 % everywhere), but because absolute value of the flux are 319 320 largest for high winds, this 7% discrepancy is also important for air-sea momentum flux values. 321 Since momentum flux is an important parameter in ocean circulation modeling, we believe 322 more research is needed (one aspects that needs more research is the fact that the newest 323 power law parameterization, A12, produces less momentum flux than all the previous ones, 324 especially in lower winds), and the parameterizations used in the models possibly need further 325 development.

326

**327**

**328 Acknowledgements**

329

We would like to express our gratitude to Ed Andreas for inspiring us. His untimely departure is an irreplaceable loss to the air-sea exchange community. We would also like to thank the entire OceanFlux team. This publication was financed with funds from Leading National Research Centre (KNOW) received by the Centre for Polar Studies for the period 2014–2018 and from OceanFlux Greenhouse Gases Evolution, a project funded by the European Space Agency, ESRIN Contract No. 4000112091/14/I-LG.

- 336 337
- 338
- 339
- 340
- 341 342
- 343

[revised manuscript text omitted]

---

## Author Comment (AC2) · 4 Oct 2018

The authors thank the reviewer for carefully reading our discussion paper and for helpful and constructive comments regarding its content and improvements. We decided to encourage our article with many conclusion both in articles, as in the case of Andreas et al., 2012, Bunker et al., 2003, and during many recent scietific conferences, that further investigation of the differences in the parameterization of the air-sea exchange coeficients are needed.

We would like to start with responding to the major comments. We contain those information also under specific comments.

The aim of the manuscript is to evaluate how much the average monthly and annually momentum transfer values depend on the choice of CD parameterizations, in other words how the selected parameterization affects the total value of momentum fluxes for large reservoirs. This allows constraining the uncertainty caused by the parameterization choice. In order to achieve this, we used observed wind field for the regions of interest, namely the North Atlantic and the European Arctic, areas where European and Americans oceanographers, including us, operate. This is where most of studies that were basis of the parameterizations we use were performed. We did some comparisons to sub-tropical basins to see the difference in uncertainty caused by the formula choice between the main study regions and less studied subtopics. In our conclusion, we do not indicate which formula should be used in the future (impossible without new data) in the NA and the EU, but the simple fact that none of the parameterizations used now is final. We don't want to suggest end users any conclusions because the differences in the parameterizations used are small, and our goal was to help them make an intelligent and deliberate decision about which parameterizations to use.

The text of the review is reproduced below in black type; our comments are in blue; and the changes in the original discussion paper are presented in italics. We reorganized the Introdaction as the reviewer sugegst, also removed equation no 7 and reorganized paragraph with this equation, changed the original title to new one and clearly state the purpose of the study. Please see the supplement material as we included all the answers and the corrected the manuscript there.

Please also note the supplement to this comment:
https://www.ocean-sci-discuss.net/os-2018-61/os-2018-61-AC2-supplement.pdf

**Figure 1.** The drag coefficient parameterization used in the study (Eqs. 7-13) as a function of wind speed $U_{10}$ (m s$^{-1}$).

[Figure]

**Fig. 1.** Figure 1

**Figure 2.** Annual mean wind speed $U_{10}$ (m s$^{-1}$) in the study area—the North Atlantic and the
European Arctic (north of the red line).

[Figure]

**Fig. 2.** Figure 2

**Figure 3.** Maps of momentum flux [N m$^{-2}$] across the sea surface (wind stress) for boreal
winters ((**a**) and (**c**)) and summers ((b) and (**d**)) for Wu (1969) and A12 drag coefficient
parameterizations (the two parameterizations with the highest and lowest average values,
respectively).

a) Wu, (1969)

[Figure]

b) Wu (1969)

**Fig. 3.** Figure 3

**Figure 4**. Monthly average momentum flux values [N m⁻²] for (**a**) global ocean, (**b**) North Atlantic, (**c**) European Arctic, and (**d**) Tropical ocean. The regions are defined in the text.

a)

[Figure]

b)

[Figure]

**Fig. 4.** Figure 4

1   **Figure 5.** Area annual average momentum flux values for (**a**) European Arctic and (**b**)
2   Tropical ocean. The vertical solid line is the average of all seven parameterizations and the
3   dashed lines are standard deviations for the presented values. Global and the North Atlantic
4   results are not shown because the relative values for different parameterizations are very
5   similar (see Table 1), scaling almost identically between the basins.
6   a)

[Figure]

**Fig. 5.** Figure 5

**Table 1.** Area average annual mean values of momentum flux (wind stress) [N m$^{-2}$] for all the studied regions and parameterizations. In each column the percentage values are normalized to A12, the parameterization that produced the smallest average flux values.

|  | Global | North Atlantic | Arctic | W. Spitsbergen | Tropics |
|---|---|---|---|---|---|
| Wu (1969) | 0.322 (114 %) | 0.330 (114 %) | 0.375 (114 %) | 0.360 (114 %) | 0.261 (119 %) |
| Garratt (1977) | 0.307 (109 %) | 0.316 (109 %) | 0.358 (109 %) | 0.344 (110 %) | 0.251 (115 %) |
| Wu (1982) | 0.311 (110 %) | 0.320 (110 %) | 0.363 (110 %) | 0.349 (111 %) | 0.255 (117 %) |
| NCEP/NCAR | 0.303 (107 %) | 0.312 (107 %) | 0.353 (107 %) | 0.341 (108 %) | 0.258 (118 %) |
| Yelland & Taylor (1996) | 0.297 (105 %) | 0.306 (105 %) | 0.348 (106 %) | 0.335 (107 %) | 0.245 (112 %) |
| Large & Yeager (2004) | 0.285 (101 %) | 0.293 (101 %) | 0.333 (101 %) | 0.320 (102 %) | 0.236 (108 %) |
| Andreas et al., (2012) | 0.283 (100 %) | 0.290 (100 %) | 0.329 (100 %) | 0.314 (100 %) | 0.219 (100 %) |

**Fig. 6.** Table 1

**Supplement:**

**RESPONSE TO REVIEWER 1 October 4, 2018**

The authors thank the reviewer for carefully reading our discussion paper and for helpful and constructive comments regarding its content and improvements. We decided to encourage our article with many conclusion both in articles, as in the case of Andreas et al., 2012, Bunker et al., 2003, and during many recent scietific conferences, that further investigation of the differences in the parameterization of the air-sea exchange coefficients are needed.

We would like to start with responding to the major comments. We contain those information also under specific comments.

The aim of the manuscript is to evaluate how much the average monthly and annually momentum transfer values depend on the choice of  $C_{\rm D}$ parameterizations, in other words how the selected parameterization affects the total value of momentum fluxes for large reservoirs. This allows constraining the uncertainty caused by the parameterization choice. In order to achieve this, we used observed wind field for the regions of interest, namely the North Atlantic and the European Arctic, areas where European and Americans oceanographers, including us, operate. This is where most of studies that were basis of the parameterizations we use were performed. We did some comparisons to subtropical basins to see the difference in uncertainty caused by the formula choice between the main study regions and less studied subtopics. In our conclusion, we do not indicate which formula should be used in the future (impossible without new data) in the NA and the EU, but the simple fact that none of the parameterizations used now is final. We don't want to suggest end users any conclusions because the differences in the parameterizations used are small, and our goal was to help them make an intelligent and deliberate decision about which parameterizations to use.

The text of the review is reproduced below in black type; our comments are in blue; and the changes in the original discussion paper are presented in italics. We reorganized the Introdaction as the reviewer suggest, also removed equation no 7 and reorganized paragraph with this equation, changed the original title to new one and clearly state the purpose of the study.

Evaluation: While each new research on  $C_D$  and  $\tau$  tries to add a bit more understating and propose improved  $C_D(U10)$  expression(s), it is hard to add something truly new after decades of investigations. Thus, it becomes important to clearly state new developments and/or new insights when publishing on this topic. In this sense, this study does not add new knowledge. It is routine. Still, I see some usefulness of this manuscript in the tabulated  $\tau$  differences for different  $C_D(U_{10})$  as this can serve as a reference to readers regarding which of many available  $C_D(U_{10})$  parameterizations to use. However, the manuscript needs much more work to be suitable for publication. In its present form, it lacks clear objective; could be better organized; and has weak conclusions. I recommend major revisions. Comments and suggestions follow.

We do our best to improve the manuscript, organize it better than it was, clearly state the objective and conclusion, also mark what new it adds to our knowledge. We conclude all of this in our respond and inside the text.

**Major comments:**

- 1) The purpose/objective of the manuscript is not clearly formulated. The title suggests two things: a climatology and a review. Two more objectives are hinted in the text (details follow). Focusing on each of these possible objectives would require quite different analyses. With the purpose not well defined, none of the possible objectives is fully developed in the manuscript. Here are specifics.
  - a) Is the objective a review of *CD* parameterizations, as the title suggests? If yes, then it is incomplete and without deep physical discussion of the progress and problems regarding parameterizing *CD*. The authors state in Line 43 that it is a must to "take into account other physical processes," yet they then focus on "wind speed parameterizations, because wind speed is a parameter that is available in every atmospheric circulation model" (Lines 68-69). The authors initiate a review of formulas by dividing them in two groups (Lines 50-55), yet, again, stop short of further discussion on different formulation of roughness length *z*0. I believe that review and physical discussion on *CD* are not the motivation of this work. After all, Andreas et al. (2012) and Edson et al. (2013, JPO, DOI: 10.1175/JPO-D-12-0173.1) provide comprehensive recent reviews of the status of parameterizing *CD*.

Yes, you are right. Our aim was not review of Cd parameterizations but checking the uncertainty caused by actual choices done by model and paper authors. That's why we used only 7 parameterizations, commonly used in literature, or in coupled circulation models. That is why we didn't do any deep physical discussion for them as the point was to study the spread between the momentum flux values for large basins with realistic wind fields. We did not try to improve the parameterizations using in-situ measurements as this was exactly what Andreas et al. did they calculations using the existing data, and we would simply reproduce their study as we did not have much additional data points. However, we could do something no one has done before, namely check the results of applications the parameterizations to realistic large-scale wind fields. In fact, we learned that no study did even calculate the average wind stress (momentum transfer) values for most ocean basins since most of the parameterizations were created. This paper meant also to fill that gap, at least partly.

Formulas have been divided into two groups to better familiarize to reader with the used parameterizations. It seems to us that their later detailed description (L73-105) is indirectly explaining the division of the formulas used into two groups.

b) Is the objective a climatology of NA and EA, as the title suggests? If yes, this motivation is not well justified and there is no analysis of the results in climatological terms. If climatology is the objective, then the authors need to tell us why they focus on the NA and EA regions? What atmospheric and oceanic conditions does the CD parameterization need to represent well in these regions? If climatology is the goal, what is the temporal or spatial reference? It seems the chosen spatial references are the global ocean and the Tropics, to which the authors compare their results for NA and EA. But if  $\tau$  is obtained globally and over many regions (Table 1), then why emphasize NA and EA in the title? Why look into differences due to CD formulation between northern regions and the Tropics, when it is certainly expected to have differences due to geography? As for a temporal reference, the authors should choose a period within or outside the 1992-2010 period which gives average atmospheric and oceanic conditions, not affected by long-term variations such as the North Atlantic Oscillation, which changes the position of jet stream and thus the wind and SST fields at the surface; these, in turn, change the wind stress. If climatology is the goal, then the authors should analyze their  $\tau$  results for trends and variations over the 1992-2010 period. Should give annual as well as inter-annual variations. Finally, in my opinion, to provide a comprehensive regional climatology, the authors should analyze long-term  $\tau$  values obtained with one, chosen CD formulation in order to clearly isolate climatologicallyrelevant variations

Our aim was not a climatology of the NA and the EA. The original title was wrongly defined and we change it. The main areas of the study: North Atlantic and European Arctic seas are the areas where European and Americans oceanographers, including us, operate. This is where most of studies that were basis of the parameterizations we use were performed. We did some comparisons to sub-tropical basins to see the difference in uncertainty caused by the formula choice between the main study regions and less studied subtopics.

c) Is the objective to evaluate *CD* parameterizations and recommend a new one for use in circulation coupled models (Line 136)? A hint for such an objective comes from the authors' conclusion "the parameterizations used

in the models possibly need upgrading" (Lines 332-333). If this is the objective, then the authors should give us a list of parameterizations used in different circulation models; discuss the advantages and limitations of these currently-used *CD* parameterizations; then demonstrate how other *CD* parameterizations would do better. For climate and circulation models, the *CD* parametrization is important for the mixing layer depth. So the authors should show how new *CD* parameterization would improve the modeling of the mixing layer. The manuscript offers limited information on what the current models use (Lines 136-140). There is no analysis on how *CD* would affect the performance of model variables related to *CD*. So the conclusion in Lines 332-333 is not convincing for modelers.

It wasn't our objective at this study/manuscript. We used the sentence ""the parameterizations used in the models possibly need upgrading" because we want to show that there is still a lot to do and there is a room for it, despite widespread statements that it is hard to add something truly new after decades of investigations. We decided to encourage our article with many conclusion both in articles, as in the case of Andreas et al., 2012, Bunker et al., 2003, and during many recent scietific conferences, that further investigation of the differences in the parameterization of the air-sea exchange coefficients are needed. However, we are agree that this sentence was not the best so we improved it: L326 used in the models possibly need further development.

d) Is the objective to demonstrate/justify the need of new measurements in NA and EA for improved *CD* parameterization in high latitudes? A hint for such an objective comes from Lines 188-189 regarding frequent ship deployment in EA, including "R/V Oceania, the ship of the institution the authors are affiliated with." If this is the objective, the manuscript would take completely different direction with discussion and analysis related to measuring methods and quality of data necessary for *CD*. Of course, this is not the objective because the authors say in Lines 65-68 that their intention "is not to re-invent or formulate a new drag parameterization … but to revisit definition of the existing drag parameterization."

The aim of the manuscript is to evaluate how much the average monthly and annually momentum transfer values depend on the choice of  $C_D$  parameterizations, in other words how the selected parameterization affects the total value of momentum fluxes for large reservoirs. This allows constraining the uncertainty caused by the parameterization choice. In order to achieve this, we used observed wind field for the regions of interest, namely the North Atlantic and the European Arctic.

As the reviewer pointed out that the aim of the manuscript was not clearly formulated, so we changed it and added properly information and corrections

**inside the text:**

L23-32 Confirming and explaining the nature and consequence of the interaction between the atmosphere and ocean is one of the great challenge in climate and sea research. These two sphere are coupled which lead to variations covering time scales from minutes to even millennia. The purpose of this article is to examine, using a modern set of software processing tools called the FluxEngine, the nature of the fluxes of momentum across the sea surface over the North Atlantic and the European Arctic. These fluxes are important to determine of current system and sea state conditions. Our goal is to evaluate how much the average monthly and annually momentum transfer values depend on the choice of  $C_D$ , using the actual wind field from the North Atlantic and the European Arctic, and demonstrate existing differences as a result of the formula used.

L134-145 In this paper we investigate how the relevant or most commonly used parameterizations for drag coefficient ( $C_D$ ) affect to value of momentum transfer values, especially in the North Atlantic (NA) and the European Arctic (EA). Our task was to demonstrate existing differences as a result of the formula used how big they can be. As is widely known, the exact equation that describes the connection between the drag coefficient and wind speed depends on the author (Geernaert, 1990). Our intention here is not to reinvent or formulate a new drag parameterization for the NA or the EA, but to revisit the existing definition of drag parameterization, and, using satellite data, to investigate how existing formulas represent the environment in the North. We concentrated on wind speed parameterizations, because wind speed is a parameter that is available in every atmospheric circulation model. Therefore, it is used in all air-sea flux parameterizations, and presently it is used even when sea state provides a closer physical coupling to the drag coefficient (for review see Geernaert et al., 1986).

The original title could cause confusion and did not clearly define the paper purpose. Therefore we change it in the revised version. The new title now is: Effect of drag coefficient formula choice on wind stress climatology in the North Atlantic and the European Arctic

e) I am listing all these possible objectives only to make the point that the authors need a well stated objective in order to focus their analysis and discussion.

I believe the authors wish to assess CD parameterizations only to decide which

one to use in some larger project.

**Yes, you are indirectly right.**

For such an assessment, the authors only need to clearly tell us why they consider the formulations (8)-(14) (i.e., no need of comprehensive review). They do not need climatology to make this assessment. One year data for  $p_i$  is enough.

We have chosen the formulations 7-13 as, in our opinion, they are most common used ones in the literature. The results between the formulae used can came from different functional Cd formulation as well as seasonal variations in NA and EA. In conclusion we add information about it.

**Differences in momentum flux mostyl came from the different functional $C_D$ formulation than from seasonal variations in the NA and the EA.**

The authors also need some reference to show them which *CD* formulation is suitable for NA and EA. Perhaps comparison of their results to regional data? Perhaps an investigation of how well a feature specific to NA and EA is represented when using different *CD* parameterizations? With such direction of the manuscript, the title may need revision to exclude claims on climatology and review.

We believe that there is no answear for that in the literature. If there was one, our paper would be pointless. We can only guess that the newer parameterizations, based on more observations, are better but this is only a guessing. Our paper tries to answer a different question: what are the differences between those parameterizations when applied to observed wind fields in a given basin. This is the question of uncertainty due to the choice of parameterization. We also do not agree that an analysis of which parameterization has lower or higher values for which winds would relly help. It is a trivial observation that those with highest values for strong winds give the highest wind stressess (with the exception of Andreas et al., which partly offsets that with low values for low winds). We added the following sentence at line 258.

L258 In the NA region with winds stronger than average for world ocean, the formula giving highest momentum transfer results are the ones with highest values for strong winds, with exception of Andreas et al. (2012) which is lower due to its low values for lower winds speeds.

2) There are several typos in the formulae that need fixing. Most importantly, it is necessary to check the coding for the calculations. These typos are as follows. In (2), U102 is in the denominator. In (7), why U10 is squared? The relationship between  $u^*$  and U10 is linear (Andresa et a., 2012, their eq. 1.10; Edson, et al., 2013, their eq. 22). In (14), need square on the wind ratio  $(u^*/U10N)2$ ; in the parenthesis, U10N2 is in the denominator; needs square on the parenthesis (compare to Andreas et a., 2012, their eq. 1.10).

We apologize for all errors in the manuscript. We checked all formulas again and corrected the mistakes.

- 3) Give better justification on choosing *CD* parameterizations (8) &(14). For example, it seems you have chosen *CD* parameterizations formulated as power law, linear, polynomial, constant. Why do you need (9) and (10)? They are so similar? Describe the merits of (11), (13) and (14), as well as their differences (e.g., data on which they are based). Do these formulations account implicitly for different processes in addition to *U*10?
- All parameterizations are important to the history of the field and all have been widely used by other authors. In our opinion, skipping any of them, especially the oldest and newest would cause the study flawed. In fact, if we had to change the number of formulas, we would rather increase it than decrease. This number was a compromise to make presenting the results graphically not too overwhelming for the reader.
- We have chosen parameterization no 9 and 10 because despite the fact that the formulas themselves are so similar and have the same source (based on Charnock's relations) Garratt in his research showed that this formula is suitable only for lights wind (over 4 m s-1), while Wu based on this statement showed that this formula can be proposed for all sea state and fit closely to the data throughout the enitre wind-velocity range. Both formulas were used in the literaturę.
- Yelland and Taylor used an automatic inertial dissipation system, over the Southern Ocean, to obtain data for wind stress estimations. During their study they examined the balance between local production and dissipitation of turbulen kinetic energy. It is the newest version of the linear parameterizations and we wanted to check (and show) how much difference in integrated momentum flux the three of them make in comparison to the other

parameterizations.

- Large and Yeager parameterizations cames from a compilation of global data sets from different sources, like NCEP/NCAR, CCSM, historical SSTand it was developed for configurations OGCM and coupled OGCM-SIM models. They used observation for winds from 1 to more than 25 m/s speed. It is used in many modern coupled circulation models.
- Andreas et al., used data from the literautre to test approach proposed by Foreman and Emeis based on the eddy-covariance flux measurement over the sea to deduc air-sea drag relations. For their study their used data for very strong winds (>24 m/s). We believe this is the most up to date parameterization and on the other hand not well known and appreciated.
- All of them used neutral-stabiility wind speed

L101-108 All of them are generated from the vertical wind profile, but they differ in the formulas used. Two of the parameterization which we chosen are formulated as power-law of the relationship between  $C_D$  and  $U_{10}$  (eq. 7 and 13), three are formulated as linear-law (eq. 8, 9 10 for light winds, and 12), and one as constant value of the relationship (eq. 11). All the above studies propose different parameterizations (see **Fig. 1**) of the drag coefficient and the function of wind speed, which reflects the difficulties in simultaneously measuring at high sea stress (or friction velocity) and wind speed.

4) Suggest re-organizing the Introduction to include Lines 37-72 plus one paragraph on why you focus on NA and EA, then another paragraph clearly stating the objective of the study. Suggest combining Lines 73-154 with Lines 195-215 in one section dedicated on *CD* parameterizations. Only parts of the historical (incomplete) review in lines 73-154 are necessary. Start with the definitions in Lines 73-93. Then introduce (8) **%**(14) one by one. Add information on MOST (Lines 115-122) and circulation models (Lines 136-140) only when they are needed, e.g., when you introduce (11) and (12), respectively. Finish the section with Lines 155-160. With this organization you will avoid the current inconsistency of presenting Fig. 1 with all parameterizations before they are described. Remove lines 122-127 and Lines 130-135 because you do not use Trenberth et al. (1989) and COARE algorithm. Unless you decide to use COARE 3.5 as a reference.

Done

L23-145 Confirming and explaining the nature and consequence of the interaction between the atmosphere and ocean is one of the great challenge in climate and sea research. These two sphere are coupled which lead to variations covering time scales from minutes to even millennia. The purpose

of this article is to examine, using a modern set of software processing tools called the FluxEngine, the nature of the fluxes of momentum across the sea surface over the North Atlantic and the European Arctic. These fluxes are important to determine of current system and sea state conditions. Our goal is revisit how the existing definition of drag parameterization affects the value of total momentum fluxes, using the actual wind field from the North Atlantic and the European Arctic and demonstrate existing differences as a result of the formula used.

The ocean surface mixed layer is a region where kinematic forcing affects the exchange of horizontal momentum and controls transport from the surface to depths (Gerbi et al., 2008, Bigdeli et al., 2017). Any attempt to properly model the momentum flux from one fluid to another as the drag force per unit area at the sea surface (surface shear stress,  $\tau$ ) take into account other physical processes responsible for generating turbulence such as boundary stress, boundary buoyancy flux, and wave breaking (Rieder et al., 1994, Jones and Toba, 2001). Fluxes across the sea surface usually depend nonlinearly on the relevant atmospheric or oceanic parameters. Over the past fifty years, as the collection of flux data has increased, many empirical formulas have been developed to express the ocean surface momentum flux as a relationship between non-dimensional drag coefficient  $(C_D)$ , wind speed  $(U_{10})$ , and surface roughness  $(z_0)$  (Wu 1969, 1982; Bunker, 1976; Garratt, 1977; Large and Pond, 1981; Trenberth et al., 1989; Yelland and Taylor, 1996, Donelan et al., 1997; Kukulka et al., 2007; Andreas et al., 2012). These formulas can be divided into two groups. One group of theories gives the  $C_D$  at level z in terms of wind speed and possibly one or more seastate parameters (for example, Geernaert et al., 1987, Yelland and Taylor, 1996, Enriquez and Friehe, 1997), while the second group provides formulas for roughness length  $z_0$  in terms of atmospheric and sea-state parameters (for example, Wu, 1969, Donelan et al., 1997, Andreas et al., 2012 (further referred to as A12)).

As the exchange of air-sea momentum is difficult to measure directly over the ocean meteorologist and oceanographers often rely on bulk formulas parameterized by Taylor (1916), that relate the fluxes to averaged wind speed through transfer coefficients:

$$\tau = \rho C_{Dz} U_z^2 \tag{1}$$

where  $\tau$  is the momentum flux of surface stress,  $\rho$  is air density,  $C_{Dz}$  is the non-dimensional drag coefficient appropriate for z height, and  $U_z$  is the average wind speed at some reference height z above the sea.  $C_{Dz}$  is commonly parameterized as a function of mean wind speed (m s-1) for neutral-stability at a 10 m reference height above mean sea level (Jones and Toba, 2001), which is identified as  $C_{DN10}$  or  $C_{D10}$  (this permits avoiding deviation for the vertical flow from the logarithmic law):

$$C_{DN10} = \frac{\tau}{\rho U_{10}^2} = (\frac{u_*}{U_{10}})^2$$
(2)

where  $u_*$  is friction velocity. Alternatively, the neutrally stratified momentum flux can be determined from the logarithmic profile, thus Eq. 1 can be express as:

$$C_{DN10} = [\kappa/\ln(10/z_0)]^2$$
(3)

where  $z_0$  (m) is the aerodynamic roughness length, which is the height, above the surface to define the measure of drag at which wind speed extrapolates to 0 on the logarithmic wind profile (Andreas et al., 2012), and  $\kappa$  is von Kármán constant ( $\kappa$ =0.4).

At the same time, we can define the value of friction velocity by the following equation:

$$\tau = \rho \, u_*^2 \tag{4}$$

Comparison with bulk formula (1) leads to the equation:

$$u_*^2 = C_{D10} U_{10}^2$$
(5)

Some of the first studies (Wu, 1969, 1982, Garrat, 1977) focused on the relationship between wind stress and sea surface roughness, as proposed by Charnock (1955), and they formulated (for winds below 15 m s-1) the logarithmic dependence of the stress coefficient on wind velocity (measured at a certain height) and the von Kármán constant. Currently common parameterizations of the drag coefficient are a linear function of 10 m wind speed ( $U_{10}$ ), and the parameters in the equation are determined empirically by fitting observational data to a curve. The general form is expressed as (Guan and Xie, 2004):

$$C_D 10^3 = (a + bU_{10}) \tag{6}$$

In this work our focus is on the fluxes of average values using seven different drag coefficient parameterizations  $(C_D)$ , chosen for their importance for the history of the field out of many published within the last half century (Bryant and Akbar, 2016).

$$10^{3} \cdot C_{D10} = 0.5U_{10}^{0.5} \qquad for \ l \ m \ s^{-l} < U_{10} < 15 \ m \ s^{-l} (Wu, 1969)$$

$$10^{3} \cdot C_{DN10} = 0.75 + 0.067U_{10} \qquad for \ 4 \ m \ s^{-l} < U < 21 \ m \ s^{-l} (B) \qquad (Garratt, 1977)$$

$$10^{3} \cdot C_{D10} = (0.8 + 0.065U_{10}) \qquad for \ U_{10} > 1 \ m \ s^{-l}$$
(9)

$$10^{3} \cdot C_{DN10} = 0.29 + \frac{3.1}{U_{10N}} + \frac{7.7}{U_{10N}^{2}} \text{ for } 3 \text{ m s}^{-l} < U_{10N} < 6 \text{ m s}^{-l} \qquad (10)$$

$$10^{3} \cdot C_{DN10} = 0.60 + 0.070U_{10N} \qquad \text{for } 6 \text{ m s}^{-l} < U_{10N} < 26 \text{ m s}^{-l} \qquad (Yelland \ and \ Taylor, 1996)$$

$$10^{3} \cdot C_{D} = 1.3 \qquad everywhere \qquad (11)$$
(NCEP/NCAR)

 $10^{3} \cdot C_{DN10} = \frac{2.7}{U_{10N}} + 0.142 + 0.076U_{10N} \qquad everywhere$ (12)

(Large and Yeager, 2004)

(Wu, 1982)

$$C_{DN10} = \left(\frac{u^*}{U_{10N}}\right)^2 = a^2 \left(1 + \frac{b}{a} U_{10N}\right)^2 \qquad everywhere$$
(13)
where
$$a = 0.0583, b = -0.243 \qquad (Andreas \ et al., 2012)$$

where  $C_{DN10}$  is the expression of neutral-stability (10-m drag coefficient),  $C_{D10}$  is the drag coefficient dependent on surface roughness,  $U_{10}$

is the mean wind speed measured at 10 m above the mean sea surface,  $U_{10N}$ is the 10-m, neutral-stability wind speed. All of them are generated from the vertical wind profile, but they differ in the formulas used. Two of the parameterization which we chosen are formulated as power-law of the relationship between  $C_D$  and  $U_{10}$  (eq. 7 and 13), three are formulated as linear-law (eq. 8, 9 10 for light winds, and 12), and one as constant value of the relationship (eq. 11). All the above studies propose different parameterizations (see **Fig. 1**) of the drag coefficient and the function of wind speed, which reflects the difficulties in simultaneously measuring at high sea stress (or friction velocity) and wind speed.

Wu (1969), based on data compiled from 12 laboratory studies and 30 oceanic observations, formulated power-law (for breezes and light winds) and linear-law (for strong winds) relationships between the wind-stress coefficient ( $C_v$ ) and wind velocity ( $U_{10}$ ) at a certain height y at various sea states. In his study, he used roughness Reynolds numbers to characterize the boundary layer flow conditions, and he assumed that the sea surface is aerodynamically smooth in the range of  $U_{10} < 3$  m s-1, transient at wind speed  $3 \text{ m s}^{-1} < U_{10} < 7 \text{ m s}^{-1}$ , and aerodynamically rough at strong winds  $U_{10} > 7$  $m s^{-1}$ . He also showed that the wind-stress coefficient and surface roughness increase with wind speed at light winds ( $U_{10} < 15 \text{ m s}^{-1}$ ) and is constant at high winds  $(U_{10} > 15 \text{ m s}^{-1})$  with aerodynamically rough flow. Garratt (1977), who assessed the 10 m neutral drag coefficient ( $C_{DN10}$ ) based on 17 publications, confirmed the previous relationship and simultaneously suggested a linear form of this relationship for light wind. Wu (1980) proposed the linear-law formula for all wind velocities and later (Wu, 1982) extended this even to hurricane wind speeds. Yelland and Taylor (1996) presented results obtained from three cruises using the inertial dissipation method in the Southern Ocean and indicate that using the linear-law relationship between the drag coefficient and wind speed (for  $U_{10} > 6 \text{ m s}^{-1}$ ) is better than using  $u_*$  with  $U_{10}$ . The NCEP/NCAR reanalysis (Kalnay et al., 1996) uses a constant drag coefficient of 1.3 x  $10^{-3}$  while, for example, the Community Climate System Model version 3 (Collins et al., 2006) uses a single mathematical formula proposed by Large and Yeager (2004) for all wind speeds. Andreas et al. (2012) based on available datasets, friction velocity coefficient versus neutral-stability wind speed at 10 m, and sea surface roughness tested the approach proposed by Foreman and Emeis (2010) for friction velocity in order to find the best fit for parameters a =

0.0583 and b = -0.243. They justify their choice by demonstrating that  $u_*$  vs.  $U_{10N}$  has smaller experimental uncertainty than  $C_{DN10}$ , and that one expression of  $C_{DN10}$  for all wind speeds overstates and overestimates results in low and high winds (**Figs.** 7 and **8** in A12).

In this paper we investigate how the relevant or most commonly used parameterizations for drag coefficient ( $C_D$ ) affect to value of momentum transfer values, especially in the North Atlantic (NA) and the European Arctic (EA). Our task was to demonstrate existing differences as a result of the formula used how big they can be. As is widely known, the exact equation that describes the connection between the drag coefficient and wind speed depends on the author (Geernaert, 1990). Our intention here is not to reinvent or formulate a new drag parameterization for the NA or the EA, but to revisit the existing definition of drag parameterization, and, using satellite data, to investigate how existing formulas represent the environment in the North. We concentrated on wind speed parameterizations, because wind speed is a parameter that is available in every atmospheric circulation model. Therefore, it is used in all air-sea flux parameterizations, and presently it is used even when sea state provides a closer physical coupling to the drag coefficient (for review see Geernaert et al., 1986).

- 5) Section 3 "Result" is straightforward. It describes Table 1, maps, and seasonal graphs. To make these results useful, you need to extend the analysis of these data, discuss what causes the differences; and suggest which *CD* parameterizations is useful for NA and EA.
- It is impossible to tell which formula is better comparing the results of its integration with wind field. That's is why we study mainly the spread (uncertainty) of result coming from the parameterization choice. However the most recent one (Andreas et al., 2012) is based on the largest measurement set, and that's why we point out what it may imply for momentum transfer, especially at low wind speed. We adds properly information inside the text.

We add properly new information inside the text at Result section:

L190-198 Despite many measurements, the drag coefficient still has wide variability at low and moderate wind speeds. Our research has showed that al lower wind values ( $<10 \text{ m s}^{-1}$ ) the differences between the drag coefficient parameterizations are greater than at higher speeds ( $> 10 \text{ m s}^{-1}$ ) and the most outlier results are those obtained from the power law parameterization

of Andreas et al., (2012), which are characterized by a sinusoidal distribution relative to the wind speed. The lower the wind speed, the higher uncertainty are, and at low winds it is uncertainty by a factor of 0.5-1.5 depending on the formula used, while at moderate winds it is uncertainty by a factor of 1.5-2.0 (Fig. 1).

**We improved also the conclusion by adding some new information.**

L288- 326 In the present work the evaluation of how the selected parameterization affects the total value of momentum fluxes for large reservoirs was assessed. This allows constraining the uncertainty caused by the parameterization choice. In order to achieve this we calculated monthly and annually average momentum fluxes using a set of software processing tools called the FluxEngine in the North Atlantic (NA) and the European Artic (EA). The NA was deifned as all sea surface areas in the Atlantic sector north of 30° N, and the EA was sea areas north of 64° N. Based on our resutls, we still do not know which one of the parameterizations can be reccomend as the most suitable for the NA and the EU study. Further investigation of the differences in the parameterization of the exchange coefficient in the various algorithm would help in resolving this problem.

Despite many measurements, the drag coefficient still has wide variability at low and moderate wind speeds. The lower the wind speed, the higher uncertainty are, and at low winds it is uncertainty by a factor of 0.5-1.5 depending on the formula used, while at moderate winds it is uncertainty by a factor of 1.5-2.0 (Fig. 1). The annual mean wind speed in the NA is 10 m s-1, and in the EU it is  $8.5 \text{ m s}^{-1}$ .

We show that the choice of drag coefficient parameterization can lead to significant differences in resultant momentum flux (or wind stress) values. Comparing the values of momentum flux across the sea surface from the power law parameterization, it showed that in both regions, with low and high winds, the parameterizations specified for all winds speeds (eq. 13) has lower values of wind stress than the parameterizations specified for light winds (eq. 7). In the Arctic, the NA, and globally the differences between the wind stress, depend on formula used, are 14 % and they are higher in low winds areas. The parameterizations generally have a decreasing trend in the resultant momentum flux values, with the most recent (Andreas et al., 2012) producing the lowest wind stress values, especially at low winds, resulting in almost 20 % differences in the tropics (Table1). The differences can be much larger on monthly scales, up to 29 % in the NA and 36 % in the EA (in

months of low winds) and even 50 % locally in the area west of Spitsbergen. For months that have the highest average winds, the percentage differences are smaller (about 7 % everywhere), but because absolute value of the flux are largest for high winds, this 7% discrepancy is also important for air-sea momentum flux values. Since momentum flux is an important parameter in ocean circulation modeling, we believe more research is needed (one aspects that needs more research is the fact that the newest power law parameterization, A12, produces less momentum flux than all the previous ones, especially in lower winds), and the parameterizations used in the models possibly need further development.

**Additional comments:**

Title: If possible (perhaps talk with the OS editor), revise the title to better reflect the purpose of your manuscript. We changed the title. Now it is:

**Effect of drag coefficient formula choice on wind stress climatology in the North Atlantic and the European Arctic**

Abstract: Too long, dilutes what you did and what you have found. Suggest substantial shortening. Avoid giving references in the abstract. Refer to different *CD* parameterizations by their specific characteristics (e.g., power law, linear, etc), not by author. Done.

Lines 18-19 and Lines 227-230: Oldest vs newest *CD* parameterization. This is not the most important difference. Frame your discussion around the functional form, the data they are based on, how well they represent low and high wind conditions.

Line 14-18: When we choose the parameterizations which increased linearly with wind speed (7-9) momentum flux were largest for all months, in compare to values from the two parameterizations which increase with wind speed sinusoidal (12 and 13) in both regions with high and low winds and  $C_D$  values were consistently higher for all wind speeds.

Line 186-190: The differences between the parameterizations are distinct (*Fig. 1*). The  $C_D$  values from the parameterizations 7-9 increased linearly

with wind speed since the results from the parameterizations 10,12,13 are characterized by sinusoidal distribution and indicating decreases for winds in the range of 0 - 10 m s-1, after which they began increase.

Line 22-23: Suggest removing this sentence. This is common sense, no need to be in the abstract.

We removed all the sentences from lines 22 to 27: For global data not much seasonal change was note due to the fact that the strongest winds are in autumn and winter as these seasons are inverse by six months for the northern and southern hemispheres. The situation was more complicated when we considered results from the North Atlantic, as the seasonal variation in wind speed is clearly marked out there. With high winter winds, the A12 parameterization was no longer the one that produces the smallest wind stress.

Line 30: "the sequence of values" is the least important thing to discuss about the differences. Discuss the physical behavior.

We removed sentences at lines 28-30: However, for low summer winds, it is the lowermost outlier. As the A12 parameterization behaves so distinctly differently with low winds, we showed seasonal results for the tropical ocean. The sequence of values for the parameterization was similar to that of the global ocean, but with visible differences betwenn NCEP/NCAR, A12 and LY04 parameterizations. Because parameterization is supported with the largest experimental data set observations of very low (or even negative) momentum flux values for developed swell and low winds, our results suggest that most circulation models overestimate momentum flux.

and reorganized the rest:

L18-21 As the one of power law parameterization (13) behaves so distinctly differently with low winds, we showed seasonal results for the tropical ocean, which were subdued for the whole region, with monthly averages in the range of 0.2 to  $0.3 \text{ N m}^2$ .

Line 76: Definition of  $\tau$  is already given in Lines 42-43. Here, and many other places, remove repeated definitions. Done.

Lines 89-90: Suggest removing this sentence, repeats definition given in Line 83.

We reorganized this sentence:

L68-69 *At the same time, we can define the value of friction velocity by the following equation:*

Line 140: I guess you mean here equation (5), which assumes proportionality; (6) modifies (5) to linear relationship.

Line 144: I think you mean here equation (7), not (8). Eq. (7) needs correction (see Major comment 2).

Line 147: Fix symbol UN10 to U10N. Check all your math symbols for correctness and consistency.

We reorganized this sentence and remove some of the information from them because it seemed unnecessary after the reorganization:

L127-133 Andreas et al. (2012) based on available datasets, friction velocity coefficient versus neutral-stability wind speed at 10 m, and sea surface roughness tested the approach proposed by Foreman and Emeis (2010) for friction velocity in order to find the best fit for parameters a = 0.0583 and b = -0.243. They justify their choice by demonstrating that  $u_*$  vs.  $U_{10N}$  has smaller experimental uncertainty than  $C_{DN10}$ , and that one expression of  $C_{DN10}$  for all wind speeds overstates and overestimates results in low and high winds (**Figs.** 7 and **8** in A12).

Line 155: Fig 1 shows parameterizations whose equations are not yet introduced. Need to introduce (8) &(14) before referring to Fig. 1. See Major comment 4.

We divided these sentences. The part was moved to lines 105-108 and part to line 133-135.

L105-108 All the above studies propose different parameterizations (see **Fig. 1**) of the drag coefficient and the function of wind speed, which reflects the difficulties in simultaneously measuring at high sea stress (or friction velocity) and wind speed.

L134-136 In this paper we investigate how the relevant or most commonly used parameterizations for drag coefficient ( $C_D$ ) affect to value of momentum transfer values, especially in the North Atlantic (NA) and the European Arctic (EA).

Line 168: Use symbol *U*10 instead of re-defining it again. Done

Line 168-169: How these data on sea roughness are used? None of your equations (8) &(14) uses sea roughness. Why then introduce these data here? Our mistake. Unnecessarily and wrongly introduced the sea surface roughness.

Line 175: Use symbol *U*10N instead of re-defining it again. Done

Lines 177-179: You do not use wave data in (8) \$(14), why do you introduce these data here?

Also our mistake. We have also introduced it unnecessarily.

Lines 180-183: Are all these details part of the FluxEngine software? Or are these done by you?

All of these details are already part of the FluxEngine. We have added relevant information to the text and reference to the literature :

L161-164 The data layers within each output file, which are details part of the FluxEngine, include statistics of the input datasets (e.g., variance of wind speed, percentage of ice cover), while the process indicator layers include fixed masks as land, open ocean, coastal classification, and ice.

Lines 186, 226, 235: Suggest re-numbering Fig. 6 to Fig. 2, then all other figures. You refer to all other figures much later in the text. Done

Lines 195-215: Need to be introduced before Line 155 (see Major comment 4). Done

Line 217: "gridded global air-sea momentum" Why global when your emphasis is on NA and EA? Is global a good reference? You need representation of average conditions (either spatially or temporally averaged) for a reference. Need to work this out.

FluxEngine software produced only global fluxes grid data and after that we calculated monthly values for separated region.

L177-180 Using the FluxEngine software, we produced global gridded monthly air-sea momentum fluxes and from these we have extracted the values for the study region, the global ocean, the NA Ocean, and its subsets: the Arctic sector of the NA and the West Spitsbergen area (WS).

Line 229: Revise "sinusoidal". The decrease at low winds is not due to sinusoidal behavior.

Line 187-190: The  $C_D$  values from the parameterizations 7 - 9 increased linearly with wind speed since the results from the parameterizations

10,12,13 are characterized by sinusoidal distribution and indicating decreases for winds in the range of 0 - 10 m s-1, after which they began increase.

Lines 246-248: Why looking into global values for seasonal variations when it is clear that opposite seasons cancel the variations? For seasonal variations, it is better to compare to Northern (or Southern) hemisphere.

We done this as we want to showed results from regionally scale against the larger background and to show the order of magnitude of differences in Northern hemisphere, and also for better detail results from regionally scale.

Line 276: "could be at statistical effect" What do you mean? Suggest revision for clarity.

We revised that to "an averaged effect". We meant that sub-tropical trade wind areas tend to have stable winds of speeds for which the Andreas et al. (2012) parameterization has almost no drag which is due to waves and wind travelling at similar velocities. We reorganized paragraph with this sentence:

L201-208 For this estimation we chose the two most-recent parameterizations (eq. 12 and 13) that showed the lowest values and change seasonally depending on the area used. As a result, these months with weak winds have significantly lower momentum flux values, which could be the effect of statistically weaker wind in ocean areas having stable winds with waves traveling in the same direction as the wind at similar speeds. Comparison showed that the A12 parameterization demonstrates almost zero sea surface drag for winds in the range of  $3 - 5 \text{ m s}^{-1}$ , which is compensated for by a certain surplus value for strong winds.

Line 286: What proportionality do you mean? Not clear.

The annual ratios of the parameterizations. We change this in the text.

L254-256 A surprising result is the annual ratios of the parameterizations values for the global, the NA, and the Arctic regions (**Fig. 4** shows that this is not true on monthly scales).

Lines 329-331: Not clear what is your conclusion here. Please revise. The sentence is now:

L319-322 For months that have the highest average winds, the percentage differences are smaller (about 7 % everywhere), but because absolute value of the flux are largest for high winds, this 7% discrepancy is also important

for air-sea momentum flux values.

Line 333: "need upgrading" From what expression? To what expression? You make all these calculations but in the end you do not recommend what is good to use. See Major comment 1c.

Changed to "need further improvements".

We do not indicate which formula should be used in the future (again impossible without new data) in the NA and the EU, but the simple fact that none of the parameterizations used now is final. We don't want End Users to draw any conclusions because the differences in the parameterizations used are small, and our goal was to help them mak an intelligent and deliberate decision about which parameterizations to use.

L322-326 Since momentum flux is an important parameter in ocean circulation modeling, we believe more research is needed (one aspects that needs more research is the fact that the newest power law parameterization, A12, produces less momentum flux than all the previous ones, especially in lower winds), and the parameterizations used in the models possibly need further development.

Line 499: Average annual mean: Area averaged? Or over the time period 1992-2010? Not clear. Please revise here and in the text.

Sorry for that, of course area average. What we had in mind in this table and fig. 5 was the average annual value of the moemntum flux divided by the surface value of each area.

We have changed it and revised in the text.

L488-490 and 514-516 **Table 1.** Area average annual mean values of momentum flux (wind stress)  $[N m^{-2}]$  for all the studied regions and parameterizations. In each column the percentage values are normalized to A12, the parameterization that produced the smallest average flux values.

L506-510 and 657-661 *Figure 5.* Area annual average momentum flux values for (**a**) European Arctic and (b) Tropical ocean. The vertical solid line is the average of all seven parameterization and the dashed lines are standard deviations for the presented values. Global and the North Atlantic results are not shown because the relative values for different parameterizations are very similar (see Table 1), scaling almost identically between the basins.

L248-256 **Table 1** and **Fig. 5** present the annual average air-sea momentum flux values (in  $N m^{-2}$ ) for all the all regions studied and all the

parameterizations. The results show that the annual North Atlantic momentum fluxes, depending on the formula used, varies from  $-0.0.290 \text{ Nm}^2$  for A12 to  $0.333 \text{ Nm}^{-2}$  for Wu (1969). In the case of global annual average, the values are -0.283 and 0.322, respectively. Table 1 shows also the same data "normalized" to the A12 data (presented as percentages of A12, which produced the lowest values for each region), which allows us to visualize the relative differences. A surprising result is the annual ratios of the parameterizations values for the global, the NA, and the Arctic regions (**Fig.** 4 shows that this is not true on monthly scales).

Figure 3a: Should show data for the Northern hemisphere if you want to use this as a reference for seasonal variations.

Yes, as in our study we compare globally data with data from Norther hemisphere. They are important as a references.

Fig. 4: Why do you need this figure? What more does it shown than Fig. 1?

We have included this figure to better illustrate the differences between the two important parameterizations. We thought about the reviewer's comment, which we agree with, therefore we diecided to remove this chart as it adds nothing new to the article. The descriptions of fig. 4 from lines 270-282 were reorganized and joined with the descriptions of fig. 1. Line 192-203

Writing style and corrections:

Line 32: I guess you mean "Because A12 parameterization.."

We removed this sentence.

Line 37: Suggest revision to read: "Wind stress at the air-sea interface influences the wind-wave interaction, including..."

Done.

Line 43: Suggest revising "must" to other word. "Must" is a firm request, which you do not follow in your subsequent considerations.

Done.

Lines 45-46: Suggest revising to read: "...fifty years, as the collection of flux data has increased, many empirical formulas..."

Done.

Line 61: Suggest revising "we chose to check" to "we investigate" or "we quantify"

Done.

Line 67: Suggest revising "accommodate" to "represent"

Done.

Lines 82-83: You have  $u^*$  in italic and non-italic. Here, and everywhere, give

mathematical symbols consistently.

Done.

Line 144: Abbreviation A12 should be introduced on first encounter, in line 50. Done.

**Effect of drag coefficient formula choice on wind stress climatology in the North Atlantic and the European Arctic**

Iwona Wrobel-Niedzwiecka, Violetta Drozdowska, Jacek Piskozub1

1Institute of Oceanology, Polish Academy of Science, ul. Powstancow Warszawy 55, 81-712 Sopot, Poland corresponding author: iwrobel@iopan.gda.pl

Key points: drag coefficient, European Arctic, North Atlantic, parameterizations

**1 Abstract**

2 In this paper we have chosen to check the differences between the relevant or most commonly 3 used parameterizations for drag coefficient  $(C_D)$  for the momentum transfer values, especially in the North Atlantic (NA) and the European Arctic (EA). We studied monthly values of air-4 5 sea momentum flux resulting from the choice of different drag coefficient parameterizations, 6 adapted them to momentum flux (wind stress) calculations using SAR wind fields, sea-ice 7 masks, as well as integrating procedures. We compared the resulting spreads in momentum flux 8 to global values and values in the tropics, an area of prevailing low winds. We show that the 9 choice of drag coefficient parameterization can lead to significant differences in resultant momentum flux (or wind stress) values. We found that the spread of results stemming from the 10 choice of drag coefficient parameterization was 14 % in the Arctic, the North Atlantic and 11 globally, but it was higher (19%) in the tropics. On monthly time scales, the differences were 12 larger at up to 29 % in the North Atlantic and 36 % in the European Arctic (in months of low 13 14 winds) and even 50 % locally (the area west of Spitsbergen). When we choose the 15 parameterizations which increased linearly with wind speed (7-9) momentum flux were largest for all months, in compare to values from the two parameterizations which increase with wind 16 speed sinusoidal (12 and 13), in both regions with high and low winds and CD values were 17 consistently higher for all wind speeds. As the one of power law parameterization (13) behaves 18 19 so distinctly differently with low winds, we showed seasonal results for the tropical ocean, which were subdued for the whole region, with monthly averages in the range of 0.2 to 0.3 N 20  $m^2$ . 21

**22 **1. Introduction**

Confirming and explaining the nature and consequence of the interaction between the 23 atmosphere and ocean is one of the great challenge in climate and sea research. These two 24 sphere are coupled which lead to variations covering time scales from minutes to even 25 millennia. The purpose of this article is to examine, using a modern set of software processing 26 tools called the FluxEngine, the nature of the fluxes of momentum across the sea surface over 27 the North Atlantic and the European Arctic. These fluxes are important to determine of current 28 system and sea state conditions. Our goal is to evaluate how much the average monthly and 29 annually momentum transfer values depend on the choice of CD, using the actual wind field 30 from the North Atlantic and the European Arctic, and demonstrate existing differences as a 31 result of the formula used. 32

33 The ocean surface mixed layer is a region where kinematic forcing affects the exchange of horizontal momentum and controls transport from the surface to depths (Gerbi et al., 2008, 34 Bigdeli et al., 2017). Any attempt to properly model the momentum flux from one fluid to 35 another as the drag force per unit area at the sea surface (surface shear stress,  $\tau$ ) take into account 36 other physical processes responsible for generating turbulence such as boundary stress, 37 38 boundary buoyancy flux, and wave breaking (Rieder et al., 1994, Jones and Toba, 2001). Fluxes across the sea surface usually depend nonlinearly on the relevant atmospheric or oceanic 39 parameters. Over the past fifty years, as the collection of flux data has increased, many 40 empirical formulas have been developed to express the ocean surface momentum flux as a 41 42 relationship between non-dimensional drag coefficient ( $C_D$ ), wind speed ( $U_{10}$ ), and surface roughness  $(z_0)$  (Wu 1969, 1982; Bunker, 1976; Garratt, 1977; Large and Pond, 1981; Trenberth 43 et al., 1989; Yelland and Taylor, 1996, Donelan et al., 1997; Kukulka et al., 2007; Andreas et 44 45 al., 2012). These formulas can be divided into two groups. One group of theories gives the  $C_D$ 46 at level z in terms of wind speed and possibly one or more sea-state parameters (for example, 47 Geernaert et al., 1987, Yelland and Taylor, 1996, Enriquez and Friehe, 1997), while the second group provides formulas for roughness length  $z_0$  in terms of atmospheric and sea-state 48 parameters (for example, Wu, 1969, Donelan et al., 1997, Andreas et al., 2012 (further referred 49 50 to as A12)).

As the exchange of air-sea momentum is difficult to measure directly over the ocean
meteorologist and oceanographers often rely on bulk formulas parameterized by Taylor (1916),
that relate the fluxes to averaged wind speed through transfer coefficients:

54
$$\tau = \rho C_{Dz} U_z^2 \tag{1}$$

55 where  $\tau$  is the momentum flux of surface stress,  $\rho$  is air density,  $C_{Dz}$  is the non-dimensional 56 drag coefficient appropriate for *z* height, and Uz is the average wind speed at some reference 57 height *z* above the sea.  $C_{Dz}$  is commonly parameterized as a function of mean wind speed (m s-1) for neutral-stability at a 10 m reference height above mean sea level (Jones and Toba, 2001), 59 which is identified as  $C_{DN10}$  or  $C_{D10}$  (this permits avoiding deviation for the vertical flow from 50 the logarithmic law):

61
$$C_{DN10} = \frac{\tau}{\rho U_{10}^2} = \left(\frac{u_*}{U_{10}}\right)^2$$
(2)

62 where  $u_*$  is friction velocity. Alternatively, the neutrally stratified momentum flux can be 63 determined from the logarithmic profile, thus Eq. 1 can be express as:

64
$$C_{DN10} = [\kappa/\ln(10/z_0)]^2$$
 (3)

where  $z_0$  (m) is the aerodynamic roughness length, which is the height, above the surface to define the measure of drag at which wind speed extrapolates to 0 on the logarithmic wind profile (Andreas et al., 2012), and  $\kappa$  is von Kármán constant ( $\kappa$ =0.4).

68 At the same time, we can define the value of friction velocity by the following equation:

$$\tau = \rho \, u_*^2 \tag{4}$$

70 Comparison with bulk formula (1) leads to the equation:

71
$$u_*^2 = C_{D10} U_{10}^2$$
 (5)

Some of the first studies (Wu, 1969, 1982, Garrat, 1977) focused on the relationship between wind stress and sea surface roughness, as proposed by Charnock (1955), and they formulated (for winds below 15 m s-1) the logarithmic dependence of the stress coefficient on wind velocity (measured at a certain height) and the von Kármán constant. Currently common parameterizations of the drag coefficient are a linear function of 10 m wind speed (U10), and the parameters in the equation are determined empirically by fitting observational data to a curve. The general form is expressed as (Guan and Xie, 2004):

$$C_D 10^3 = (a + bU_{10}) \tag{6}$$

80 In this work our focus is on the fluxes of average values using seven different drag 81 coefficient parameterizations ( $C_D$ ), chosen for their importance for the history of the field out 82 of many published within the last half century (Bryant and Akbar, 2016).

83
$$10^3 \cdot C_{D10} = 0.5U_{10}^{0.5}$$
 for  $1 \text{ m s}^{-1} < U_{10} < 15 \text{ m s}^{-1}$  (7)
84 (Wu, 1969)

85
$$10^3 \cdot C_{DN10} = 0.75 + 0.067 U_{10}$$
 for 4 m s-1 < U < 21 m s-1 (8)
86 (Garratt,

87 1977)

79

88
$$10^3 \cdot C_{D10} = (0.8 + 0.065U_{10})$$
 for  $U_{10} > 1 \text{ m s}^{-1}$  (9)
89 (Wu, 1982)

90
$$10^3 \cdot C_{DN10} = 0.29 + \frac{3.1}{U_{10N}} + \frac{7.7}{U_{10N}^2}$$
 for 3 m s-1 <  $U_{10N} < 6$  m s-1 (10)
103 ·  $C_{DN10} = 0.60 + 0.070 U_{10N}$  for 6 m s-1 <  $U_{L} < 26$  m s-1

91
$$10^{\circ} \cdot c_{DN10} = 0.60 + 0.070 U_{10N}$$
 for 6 m s <  $U_{10N} < 26$  m s
92 (Yelland and Taylor, 1996)

93
$$10^3 \cdot C_D = 1.3$$
 everywhere (11)
94 (NCEP/NCAR)

95
$$10^3 \cdot C_{DN10} = \frac{2.7}{U_{10N}} + 0.142 + 0.076U_{10N}$$
 everywhere
96 (L

(Large and Yeager, 2004)

(12)

97
$$C_{DN10} = \left(\frac{u^*}{U_{10N}}\right)^2 = a^2 \left(1 + \frac{b}{a} U_{10N}\right)^2$$
 everywhere (13)
98 where  $a = 0.0583, b = -0.243$  (Andreas et al., 2012)

where  $C_{DN10}$  is the expression of neutral-stability (10-m drag coefficient),  $C_{D10}$  is the drag 99 coefficient dependent on surface roughness,  $U_{10}$  is the mean wind speed measured at 10 m above 100 the mean sea surface, U10N is the 10-m, neutral-stability wind speed. All of them are generated 101 102 from the vertical wind profile, but they differ in the formulas used. Two of the parameterization 103 which we chosen are formulated as power-law of the relationship between  $C_D$  and  $U_{10}$  (eq. 7) and 13), three are formulated as linear-law (eq. 8, 9 10 for light winds, and 12), and one as 104 105 constant value of the relationship (eq. 11). All the above studies propose different parameterizations (see Fig. 1) of the drag coefficient and the function of wind speed, which 106

reflects the difficulties in simultaneously measuring at high sea stress (or friction velocity) andwind speed.

[revised manuscript text omitted]

**3. Results and Discussion**

Using the FluxEngine software, we produced global gridded monthly air-sea 177 momentum fluxes and from these we have extracted the values for the study region, the global 178 179 ocean, the NA Ocean, and its subsets: the Arctic sector of the NA and the West Spitsbergen 180 area (WS). Some of the parameterizations used were limited to a restricted wind speed domain. 181 We used them for all the global wind speed data to avoid data gaps for winds that were too high 182 or too low for a given parameterization (Fig. 1). However, circulation models have the very same constraint and, therefore, the procedure we used emulated using the parameterization in 183 184 oceanographic and climate modeling.

185 Since wind velocity was used to estimate  $C_D$ , **Fig. 1** shows a wide range of empirical 186 formulas and **Fig. 2** shows annual mean wind speed  $U_{10}$  (m s-1) in the NA and the EA. The 187 differences between the parameterizations are distinct (**Fig. 1**). The  $C_D$  values from the

parameterizations 7 - 9 increased linearly with wind speed since the results from the 188 parameterizations 10,12,13 are characterized by sinusoidal distribution and indicating 189 decreases for winds in the range of 0 - 10 m s-1, after which they began increase. Despite many 190 measurements, the drag coefficient still has wide variability at low and moderate wind speeds. 191 Our research has showed that al lower wind values ( $<10 \text{ m s}^{-1}$ ) the differences between the drag 192 coefficient parameterizations are greater than at higher speeds (> 10 m s-1) and the most outlier 193 results are those obtained from the power law parameterization of Andreas et al., (2012). The 194 195 lower the wind speed, the higher uncertainty are, and at low winds it is uncertainty by a factor of 0.5-1.5 depending on the formula used, while at moderate winds it is uncertainty by a factor 196 of 1.5-2.0 (Fig. 1). At a wind value of about 15 m s-1, the results from eq. 8, 9, and 13 overlapped 197 providing the same values for the drag coefficient parameterizations. Additionally, we 198 199 compared directly the results of the two parameterizations for the drag air-sea relation that uses 200 different dependencies (Fig. 1). For this estimation we chose the two most-recent parameterizations (eq. 12 and 13) that showed the lowest values and change seasonally 201 202 depending on the area used. As a result, these months with weak winds have significantly lower 203 momentum flux values, which could be the effect of statistically weaker wind in ocean areas having stable winds with waves traveling in the same direction as the wind at similar speeds. 204 205 Comparison showed that the A12 parameterization demonstrates almost zero sea surface drag for winds in the range of 3 - 5 m s-1, which is compensated for by a certain surplus value for 206 strong winds. The small drag coefficient values facilitate what Grachev and Fairall (2001) 207 describe as the transfer of momentum from the ocean to the atmosphere at wind speeds of 2 - 4 208 m s-1, which correspond to the negative drag coefficient value. Such events require specific 209 meteorologist conditions, but this strongly suggests that the average  $C_D$  value for similar wind 210 speeds could be close to zero. The annual mean wind speed in the NA is 10 m s-1, and in the 211 EA it is  $8.5 \text{ m s}^{-1}$  (Fig. 2). 212

Figure 3 presents maps of the mean boreal winter DJF and summer JJA momentum 213 fluxes for the chosen  $C_D$  parameterizations (Wu, 1969 and A12 – the ones with the largest and 214 smallest  $C_D$  values). The supplementary materials contain complete maps of annual and 215 seasonal means for all the parameterizations. The zones of the strongest winds are in the extra-216 217 tropics in the winter hemisphere (southern for JJA and northern for DJF). The older Wu (1969) 218 parameterization produces higher wind stress values than A12 in both regions with high and low winds and  $C_D$  values are consistently higher for all wind speeds except the lowest ones 219 (which, after multiplying by  $U^2$ , produced negligible differences in wind stress for the lowest 220 winds). The average monthly values for each of the studied areas are shown in Fig. 4. Generally, 221 this illustrates that the sinusoidal the drag coefficient parameterization is, the smaller the 222 223 calculated momentum flux is. For global data (Fig. 4a), not much seasonal change is noted, 224 because the strongest winds are in fall and winter, but these seasons are the opposite in the northern and southern hemispheres. The parameterization with the largest momentum flux 225 values for all months is that of Wu (1969), the linear one, while the two parameterizations with 226 the lowest values are the sinusoidal ones (Large and Yeager, 2004 and A12). For the NA (Fig. 227 4b), with is much more pronounced seasonal wind changes, the situation is more complicated. 228 229 With high winter winds, the A12 parameterization is no longer the one that produces the smallest wind stress (it is actually in the middle of the seven). However, for low summer winds, 230

it is the lowermost outlier. Actually, in summer, the constant  $C_D$  value used by the NCEP/NCAR 231 reanalysis produces the highest wind stress values in the NA. The situation is similar for the 232 233 EA (a subset of the NA), the wind stress values of which are shown in Fig. 4c, and for the WS 234 area (not show). In the Arctic summer, A12 produces the least wind stresses, while all the other parameterizations look very similar qualitatively (even more so in the Arctic than in the whole 235 NA). Because the A12 parameterization behaves so distinctly differently with low winds, we 236 237 also show seasonal results for the tropical ocean (Fig. 4d). The seasonal changes are subdued for the whole tropical ocean with the slight domination of the Southern Hemisphere (the 238 strongest winds are during the boreal summer) with generally lower momentum transfer values 239 (monthly averages in the range of 0.2 to 0.3 N m-2 compared to 0.2 to 0.4 N m-2 for the NA and 240 0.2 to 0.5 N m-2 for the Arctic). The sequence of values for the parameterization is similar to 241 that of the global ocean, but there are differences. Here the NCEP/NCAR constant 242 parameterization is the second highest (instead of Wu, 1982 for the global ocean) while, unlike 243 in the case of the global ocean, A12 produces visibly lower values than does the Large and 244 245 Yeager (2004) parameterization.

246 Table 1 and Fig. 5 present the annual average air-sea momentum flux values (in N m 247  $^{2}$ ) for all the all regions studied and all the parameterizations. The results show that the annual North Atlantic momentum fluxes, depending on the formula used, varies from -0.0.290 N m-2 248 for A12 to 0.333 N m-2 for Wu (1969). In the case of global annual average, the values are -249 0.283 and 0.322, respectively. Table 1 shows also the same data "normalized" to the A12 data 250 251 (presented as percentages of A12, which produced the lowest values for each region), which 252 allows us to visualize the relative differences. A surprising result is the annual ratios of the 253 parameterizations values for the global, the NA, and the Arctic regions (Fig. 4 shows that this is not true on monthly scales). The spread of the momentum flux results is 14 % in all three 254 255 regions, and even flux values themselves are larger in the NA than globally and larger in the Arctic than in the whole of the NA basin. In the NA region with winds stronger than average 256 for world ocean, the formula giving highest momentum transfer results are the ones with highest 257 values for strong winds, with exception of Andreas et al. (2012) which is lower due to its low 258 259 values for lower winds speeds. The smaller WS region, with winds that are, on average, weaker 260 than those of the whole Arctic (but stronger than those of the whole NA), had slightly different ratios of the resultant fluxes. For the tropical ocean, which is included for comparison because 261 of its weaker winds, the spread in momentum flux values on an annual scale is 19 %. The 262 spreads are even larger on monthly scales (not shown). The difference between A12 and Wu 263 264 (1969) and NCEP/NCAR (the two parameterizations producing the largest fluxes on monthly scales) are 27 % and 29 % for the NA (in July), 31 % and 36 % for the Arctic (in June), 42 % 265 and 51 % for the WS region (in July) and 23 % and 22 % for the tropical ocean (in April), 266 respectively. Seasonality in the tropics is weak, therefore, the smallest monthly difference of 267 268 16 % (July) is larger than the difference for the global data in any month (the global differences between the parameterizations have practically no seasonality). On the other hand, the smallest 269 270 monthly differences between the parameterizations in the NA, the Arctic, and the WS regions are all 7 %, in the month of the strongest winds (January). 271

272 Because the value of momentum flux is important for ocean circulation, its correct calculation in coupled models is very important, especially in the Arctic, where cold halocline 273 274 stratification depends on the amount of mixing (Fer, 2009). We show that with the 275 parameterization used in modelling, such as the NCEP/NCAR constant parameterization and Large and Yeager (2004), production stress results differ by about 5 %, on average (both in the 276 277 Arctic and globally), and the whole range of parameterizations leads to results that differ, on 278 average, by 14 % (more in low wind areas) and much more on monthly scales. One aspect that 279 needs more research is the fact that the newest parameterization, A12, produces less momentum flux than all the previous ones, especially in lower winds (which, by the way, continues the 280 trend of decreasing values throughout the history of the formulas discussed). The A12 281 282 parameterization is based on the largest set of measurements of friction velocity as a function 283 of wind speed and utilizes the recently discovered fact that b in equation (7) is not negligible. It also fits the observations that developed swell at low wind velocity has celerity which leads 284 285 to zero or even negative momentum transfer (Grachev and Fairall, 2001). Therefore, the 286 significantly lower A12 results for the tropical ocean (the trade wind region) and months of low 287 winds elsewhere could mean that most momentum transfer calculations are overestimated. This 288 matter needs further study, preferably with new empirical datasets.

**289 **4.** Conclusions**

290 In the present work the evaluation of how the selected parameterization affects the total 291 value of momentum fluxes for large reservoirs was assessed. This allows constraining the 292 uncertainty caused by the parameterization choice. In order to achieve this we calculated 293 monthly and annually average momentum fluxes using a set of software processing tools called 294 the FluxEngine in the North Atlantic (NA) and the European Artic (EA). The NA was defined 295 as all sea surface areas in the Atlantic sector north of 30° N, and the EA was sea areas north of 64° N. Based on our results, we still do not know which one of the parameterizations can be 296 297 reccomend as the most suitable for the NA and the EU study. Further investigation of the 298 differences in the parameterization of the exchange coefficient in the various algorithm would 299 help in resolving this problem.

Bespite many measurements, the drag coefficient still has wide variability at low and moderate wind speeds. The lower the wind speed, the higher uncertainty are, and at low winds it is uncertainty by a factor of 0.5-1.5 depending on the formula used, while at moderate winds it is uncertainty by a factor of 1.5-2.0 (Fig. 1). The annual mean wind speed in the NA is 10 m  $s^{-1}$ , and in the EU it is 8.5 m s-1.

305 We show that the choice of drag coefficient parameterization can lead to significant differences in resultant momentum flux (or wind stress) values. Comparing the values of 306 307 momentum flux across the sea surface from the power law parameterization, it showed that in 308 both regions, with low and high winds, the parameterizations specified for all winds speeds (eq. 309 13) has lower values of wind stress than the parameterizations specified for light winds (eq. 7). In the Arctic, the NA, and globally the differences between the wind stress, depend on formula 310 311 used, are 14 % and they are higher in low winds areas. The parameterizations generally have a decreasing trend in the resultant momentum flux values, with the most recent (Andreas et al., 312 313 2012) producing the lowest wind stress values, especially at low winds, resulting in almost 20

% differences in the tropics (Table1). The differences can be much larger on monthly scales, 314 up to 29 % in the NA and 36 % in the EA (in months of low winds) and even 50 % locally in 315 the area west of Spitsbergen. For months that have the highest average winds, the percentage 316 differences are smaller (about 7 % everywhere), but because absolute value of the flux are 317 318 largest for high winds, this 7% discrepancy is also important for air-sea momentum flux values. 319 Since momentum flux is an important parameter in ocean circulation modeling, we believe 320 more research is needed (one aspects that needs more research is the fact that the newest 321 power law parameterization, A12, produces less momentum flux than all the previous ones, 322 especially in lower winds), and the parameterizations used in the models possibly need further 323 development.

324

**325**

**326 Acknowledgements**

327

We would like to express our gratitude to Ed Andreas for inspiring us. His untimely departure is an irreplaceable loss to the air-sea exchange community. We would also like to thank the entire OceanFlux team. This publication was financed with funds from Leading National Research Centre (KNOW) received by the Centre for Polar Studies for the period 2014–2018 and from OceanFlux Greenhouse Gases Evolution, a project funded by the European Space Agency, ESRIN Contract No. 4000112091/14/I-LG.

- 334 335
- 336
- 337
- 338
- 339
- 340 341

[revised manuscript text omitted]

- 529